# RETHINKING PREDICTIVE LLM ROUTING: WHEN SIMPLE KNN BEATS COMPLEX LEARNED ROUTERS

## ABSTRACT

As large language models (LLMs) grow in scale and specialization, routing—selecting the best model for a given input—has become essential for efficient and effective deployment. While recent methods rely on increasingly complex learned routing strategies, their dependence on disparate training data and evaluation setups makes comparison and generalization difficult. In this work, we fundamentally rethink LLM routing by questioning whether such complexity is necessary. We show that a well-tuned k-Nearest Neighbors (kNN) approach not only matches but often outperforms state-of-the-art learned routers while being significantly more efficient. To support systematic evaluation, we introduce a suite of standardized routing benchmarks spanning instruction-following, question-answering, and reasoning tasks, as well as the first multi-modal routing dataset involving visual inputs. Our theoretical analysis reveals that the strong locality properties of model performance in embedding space enable simple non-parametric methods to achieve superior routing decisions with lower sample complexity than parametric approaches. These findings challenge the prevailing trend toward sophisticated architectures and demonstrate that simple, interpretable approaches can be surprisingly effective for LLM routing.

## 1 INTRODUCTION

The proliferation of large language models (LLMs) in recent years has created an increasingly diverse ecosystem of models with varying sizes, capabilities, and specializations (Team et al., 2023; Jaech et al., 2024; Guo et al., 2025; Yang et al., 2024). As organizations and users gain access to this expanding array of models—each with different strengths, computational demands, and cost profiles—a crucial challenge has emerged: how to intelligently select the most appropriate model for a given input. This challenge, known as LLM routing, carries significant implications for both cost-effective deployment and optimal user experience (Varangot-Reille et al., 2025; Chen et al., 2025).

Current LLM routing approaches typically employ sophisticated learned policies that leverage various signals, such as model preferences (Ong et al., 2024), prompt embeddings (Chen et al., 2024), or external scoring functions (Lu et al., 2023; Stripelis et al., 2024). These methods vary in their implementation strategies—some frame routing as a selection or classification problem (Stripelis et al., 2024; Ding et al., 2024; Feng et al., 2024), while others develop predictive models to estimate utility scores of each LLM for specific inputs (Nguyen et al., 2024; Li, 2025). The field has witnessed an escalating trend toward increasingly complex architectures, including graph neural networks, attention mechanisms, and multi-layered predictive models, often without rigorous comparison to simple baselines. Moreover, these diverse approaches rely on different training datasets, evaluation protocols, and underlying assumptions, creating challenges for fair comparisons and raising questions about their generalizability.

While the field continues to develop increasingly complex routing solutions, we fundamentally question whether such sophistication is necessary. In this paper, we take a step back and reconsider LLM routing from first principles, posing a critical question: *how far can we get with simple, non-parametric methods like k-Nearest Neighbors (kNN)?* Our investigation yields a surprising finding: when carefully implemented and tuned, a simple kNN-based router not only matches but often

outperforms a wide range of complex learned approaches across diverse routing scenarios, while offering substantial advantages in computational efficiency and robustness.

To enable systematic evaluation and address the inconsistent protocols that have hindered meaningful comparisons, we introduce a comprehensive benchmark suite for LLM routing. This suite spans instruction-following, question-answering, and reasoning tasks, establishing consistent evaluation protocols and performance metrics. We further extend our investigation to multi-modal scenarios by developing the first benchmark for routing between vision-language models, demonstrating that the effectiveness of simple approaches generalizes beyond text-only applications.

Our comprehensive evaluation reveals several key insights that challenge prevailing assumptions in the field. First, kNN-based routing achieves competitive or superior performance while being significantly more computationally efficient than complex alternatives. Second, simple methods demonstrate greater robustness under distribution shift, maintaining more stable performance when applied to out-of-distribution queries. Third, through theoretical analysis, we show that the strong locality properties in embedding spaces—where semantically similar queries benefit from similar models—enable non-parametric methods to achieve effective routing with lower sample complexity than parametric approaches.

These findings represent more than just an empirical comparison; they constitute a fundamental rethinking of complexity assumptions in LLM routing. Our work demonstrates that the field may be over-engineering solutions to a problem that can be effectively addressed with simpler, more interpretable approaches. This has important practical implications: simpler routing methods can significantly reduce the computational and engineering overhead required to deploy sophisticated multi-model systems, potentially democratizing access to effective LLM routing for organizations with limited resources.

By challenging the prevailing trend toward architectural sophistication, our work redirects research attention toward understanding when and why simple methods suffice, providing both practical guidance for practitioners and theoretical insights that can inform future routing system design. We believe this represents an important contribution to the field's maturation, emphasizing the value of thorough baseline evaluation before investing in complex solutions.

## 2 BACKGROUND AND RELATED WORKS

As the ecosystem of large language models (LLMs) becomes increasingly diverse, optimizing the trade-off between performance and computational cost has become a central research challenge. To address this challenge, three primary strategies have emerged: ensemble methods, cascading approaches, and routing systems.

**Ensemble methods** improve robustness and answer quality by aggregating outputs from multiple LLMs. Prior works such as LLM-Blender (Jiang et al., 2023), Blending (Lu et al., 2024), and Fusion (Wang et al., 2023) demonstrate that ensembling can yield strong performance across a range of tasks. However, this approach comes at a significant cost—ensemble methods require simultaneous inference from multiple models, leading to increased latency and computational overhead that scales linearly with the number of models.

**Cascading approaches** aim to reduce these costs by invoking models in a sequence of increasing capability and expense. Systems like FrugalGPT (Chen et al., 2023), AutoMix (Aggarwal et al., 2024), and Two-tier Selection (Ramírez et al., 2024) start with smaller, faster models and escalate to more capable ones only when necessary. While this sequential design can lower the average cost compared to always using the most expensive model, it still incurs multiple model calls for challenging queries and often relies on auxiliary quality estimation mechanisms, which can introduce additional latency and system complexity.

**Routing systems** offer the most direct and efficient alternative by selecting a single LLM to handle each query. These systems eliminate the need for multiple model calls, minimizing both cost and latency while maintaining the flexibility to choose the most appropriate model for each input. Most routing approaches rely on performance prediction to guide model selection. Some methods predict evaluation scores or reward values for each model given an input (Shnitzer et al., 2023; Hari & Thomson, 2023; Šakota et al., 2024), while others take a comparative approach by estimating win rates or preference relationships between model pairs (Ding et al., 2024; Ong et al., 2024).

Table 1: Overview of existing routers.

| Routers | Routing Formulation | Training Signal | Training Objective | Support Set |
|---|---|---|---|---|
| TensorOpera (Stripelis et al., 2024) | Classification | BERTsim score | Cross Entropy | No |
| HybridLLM (Ding et al., 2024) | Classification | BART score | Cross Entropy | No |
| ZOOTER (Lu et al., 2023) | Classification | Reward model | KL divergence | No |
| GraphRouter (Feng et al., 2024) | Classification | Evaluation metric | Cross Entropy | Yes |
| RouterDC (Chen et al., 2024) | Embedding Similarity | Answer correctness | Contrastive learning | No |
| MetaLLM (Nguyen et al., 2024) | Multi-armed Bandit | Online utility score | Linear reward model | No |
| LLMBandit (Li, 2025) | Multi-armed Bandit | Online utility score | PPO policy learning | No |
| RouteLLM (Ong et al., 2024) | Ranking | Pairwise preference | Contrastive loss | No |
| Eagle (Zhao et al., 2024) | Ranking | Pairwise preference | ELO calculation | Yes |
| Routoo (Mohammadshahi et al., 2024) | Utility Prediction | Answer correctness | Cross Entropy | No |
| FORC (Šakota et al., 2024) | Utility Prediction | Evaluation metric | Cross Entropy | No |
| Tryage (Hari & Thomson, 2023) | Utility Prediction | Expert model losses | Divergence | No |

More recently, MetaLLM (Nguyen et al., 2024) and LLMBandit (Li, 2025) frame the problem as a contextual bandit, learning routing policies that balance exploration and exploitation for model selection.

Despite the diversity of routing approaches, the field has increasingly gravitated toward complex architectures—including graph neural networks, attention mechanisms, and sophisticated predictive models—often without systematic comparison to simpler alternatives. Table 1 provides an overview of existing routing approaches, categorized by their formulation, training signals, and whether they utilize support sets, illustrating this trend toward complexity.

In this work, we focus on the routing setting due to its strong efficiency potential and practical relevance. Our study systematically evaluates both simple and complex routing methods, revealing the surprising effectiveness of non-parametric approaches and raising important questions about when architectural complexity is truly necessary for effective LLM routing.

## 3  PREDICTIVE LLM ROUTING

The goal of LLM routing is to select the most appropriate model from a set of available LLMs to process a given input query, subject to constraints such as performance, cost, or latency. Formally, let $x$ denote a query and $\mathcal{M} = \{m_1, m_2, \ldots, m_M\}$ represent the pool of candidate models. For each $(x, m)$ pair, we define an unknown performance score $s(x, m)$ that captures the quality of model $m$'s response to query $x$ (e.g., accuracy or reward), and a cost function $c(x, m)$ that reflects the resource cost of invoking model $m$ on input $x$ (e.g., latency or compute cost, which may depend on input and response length). The objective is to select a model that maximizes utility while balancing cost:

$$m^* = \arg \max_{m \in \mathcal{M}} s(x, m) - \lambda \cdot c(x, m), \tag{1}$$

where $\lambda$ is a user-defined trade-off parameter governing the preference for performance versus cost.

We study two complementary classes of routing approaches, drawing inspiration from reinforcement learning (RL): **model selection policies**, which learn a routing policy to directly predict the optimal model, and **utility prediction methods**, which estimate the score $s(x, m)$ and cost $c(x, m)$ for each model and select the one with the highest predicted utility. Table 1 provides an overview of existing routers mapped to these categories.

**Routing as LLM Selection** In this formulation, routing is framed as a direct model selection problem: given an input $x$, the router learns a policy $\pi(x)$ that maps queries to models, aiming to predict the optimal choice $m^*$. This perspective is analogous to policy gradient methods in RL, where an agent learns to output actions directly without explicitly estimating the value of all alternatives.

The policy $\pi(x)$ can be parametrized in various ways. Several approaches structure it as a classifier, with gold routing labels derived from preference data (Ong et al., 2024), reward scores (Stripelis et al., 2024), or thresholded evaluation metrics (Ding et al., 2024). Other methods parametrize the policy as a distance measure between query and model embeddings, formulating the problem as an embedding learning task trained using contrastive objectives (Chen et al., 2024). A third line of work frames routing as a contextual bandit problem, directly learning routing policies from online feedback (Nguyen et al., 2024; Li, 2025). These approaches differ in their learning objectives and adaptation capabilities, but all ultimately aim to map inputs to optimal model selections.

**Routing as Utility Prediction** Alternatively, routing can be framed as a utility estimation problem: for each input $x$ and candidate model $m$, the system predicts a scalar utility score $\hat{u}(x, m) = \hat{s}(x, m) - \lambda \cdot \hat{c}(x, m)$, where $\hat{s}(x, m)$ and $\hat{c}(x, m)$ are predicted score and cost. Routing is then performed by selecting the model with the highest predicted utility.

This formulation directly parallels $Q$-learning in reinforcement learning, where the $Q$-function $Q(x, a)$ estimates the expected return of taking action $a$ in state $x$, and the policy selects the action with the highest $Q$-value. In the LLM routing context, "actions" correspond to candidate models, and utility predictors serve as the $Q$-function. This approach enables more nuanced reasoning over model selection and naturally accommodates complex scenarios involving multi-objective tradeoffs. By explicitly estimating both performance and cost dimensions, utility prediction provides a flexible framework for balancing quality and efficiency in diverse deployment settings.

**Leveraging Support Sets for Enhanced Routing** Beyond direct policy learning and utility prediction, routing can be enhanced by considering each query within its neighborhood context. In this approach, a query $x$ is routed by leveraging a support set $\mathcal{D}_{\text{support}} = \{(x_i, m, s(x_i, m), c(x_i, m))\}$ that captures performance scores and computational costs for semantically similar prompts $x_i$, providing valuable contextual signals about how different models perform on related inputs.

Both the model selection and utility prediction frameworks can incorporate support sets. Non-parametric methods like $k$-Nearest Neighbors (kNN) estimate $s(x, m)$ and $c(x, m)$ by retrieving similar inputs $x_i$ from $\mathcal{D}_{\text{support}}$ and aggregating their recorded outcomes. Meanwhile, parametric routers can be designed to process both the target query and its neighbors, enabling them to learn from local performance-cost landscapes rather than treating each query in isolation.

This contextual approach bridges the gap between instance-level prediction and dataset-level routing patterns, enabling practical test-time adaptive routing that can respond to distribution shifts without requiring complete retraining. The effectiveness of this approach depends critically on the locality properties of model performance in the embedding space—if semantically similar queries tend to benefit from similar models, then neighborhood-based methods can achieve strong routing performance. As we demonstrate in our experiments, this locality assumption holds remarkably well in practice, enabling even simple kNN-based methods to achieve competitive or superior performance compared to sophisticated parametric approaches.

## 4 ROUTING BENCHMARKS

To enable systematic evaluation of routing approaches, we develop standardized benchmarks that address the inconsistent evaluation protocols which have hindered meaningful comparisons between routing methods.

### 4.1 TEXT-BASED ROUTING BENCHMARKS

While RouterBench (Hu et al., 2024) provides a valuable starting point for routing evaluation with 11 LLMs across 6 tasks, its limited model pool constrains applicability to diverse real-world scenarios. To address this limitation, we construct a more comprehensive benchmark incorporating a broader range of models and tasks from established evaluation frameworks.

Our benchmark leverages three widely-used LLM evaluation leaderboards: AlpacaEval (Dubois et al., 2024), Open LLM Leaderboard v2 (Fourrier et al., 2024), and HELM-Lite (Liang et al., 2022). From each leaderboard, we select three model families (e.g., OpenAI GPT series, Google Gemini series), with multiple variants per family to represent practical routing scenarios. The performance scores $s(x, m)$ are derived directly from the evaluation metrics reported in each respective leaderboard, while the costs $c(x, m)$ are calculated using the actual pricing of each API, based on input and output token counts. For detailed benchmark construction, model selection criteria, and scoring methodology, please refer to Appendix B.

### 4.2 VISION-LANGUAGE ROUTING BENCHMARKS

To address routing challenges in the increasingly important multi-modal domain, we introduce the first benchmark for routing between vision-language models. Our benchmark builds upon vHELM

(Lee et al., 2024), a comprehensive evaluation framework that systematically assesses capabilities across visual understanding, reasoning, and instruction following.

We incorporate leading multi-modal models from Claude and OpenAI's families, selecting variants that represent different performance-cost tradeoffs. For evaluation, we curate five diverse datasets focusing on visual question answering and visual reasoning tasks, carefully chosen to represent varying levels of complexity and different visual understanding requirements. Detailed information about model selection, dataset characteristics, and evaluation metrics is provided in Appendix B.

This benchmark enables evaluation of how routing approaches handle multi-modal inputs, where optimal model selection depends not only on the text query but also on visual content characteristics, image quality, and the specific interplay between visual and textual components of the task.

### 4.3 EVALUATION PROTOCOL

To systematically evaluate routing approaches with different objectives, we develop distinct evaluation protocols for utility prediction and model selection approaches:

**Utility Prediction Evaluation (Recommended)** These methods explicitly predict performance scores and costs, allowing us to trace the complete Pareto front by varying the trade-off parameter $\lambda$ across a wide range. We assess router effectiveness by measuring the area under the curve (AUC) of the non-decreasing convex hull in the cost-performance space. This metric comprehensively captures a router's ability to balance performance and cost across the entire spectrum of preference settings, with higher AUC indicating better overall routing decisions. We normalize score and cost values so that the maximum AUC score is 100.

**Selection-Based Evaluation** These methods directly map queries to models without explicit utility scores, making construction of a full Pareto front challenging. We evaluate these routers at three distinct cost-performance preferences: low-cost ($\lambda = 1.0/c_{\max}$), balanced ($\lambda = 0.5/c_{\max}$), and high-performance ($\lambda = 0.1/c_{\max}$), where $c_{\max}$ is the maximum cost in each benchmark.

However, this approach has important limitations. Since selection-based methods only report accuracy under fixed preference settings, they can obscure true cost-performance trade-offs. High accuracy scores may simply reflect selection of expensive models rather than intelligent routing decisions, as preference parameters only indirectly control cost through the utility function.

For both evaluation approaches, we report performance relative to oracle and random baselines, enabling precise quantification of routing effectiveness. Given the limitations of selection-based routing formulation, our main results focus on utility prediction evaluation, with selection-based results provided in Appendix D for completeness.

## 5 ROUTING APPROACHES

Building on our benchmarking framework, we evaluate routing approaches ranging from simple non-parametric methods to sophisticated neural architectures. We organize these approaches by complexity, with all methods supporting both utility prediction (estimating performance scores and costs) and model selection (directly classifying queries to models) formulations using consistent input representations.

**Non-Parametric Methods** **k-Nearest Neighbors (kNN)** routes queries by retrieving the $k$ most similar examples from a training set and aggregating their performance outcomes. For utility prediction, it averages observed scores and costs from neighbors; for model selection, it uses majority voting among top-performing models for similar queries. This approach requires no training and adapts to new queries by leveraging local neighborhood information.

**Linear Methods** The **Linear Router** employs linear regression (utility prediction) or logistic regression (model selection) over query embeddings, capturing linear relationships between query features and routing outcomes. The **Linear Matrix Factorization (MF) Router**, inspired by RouteLLM (Ong et al., 2024), assigns learnable embeddings to each model and predicts utility scores through linear operations on prompt-model embedding interactions, explicitly modeling the relationship between query characteristics and model capabilities.

Table 2: AUC scores on a range of text routing benchmarks. All methods predict utility scores to inform routing decisions. Higher is better.

| | AlpacaEval | | | HELM-Lite | | | OpenLLM | | | RouterBench | Avg |
|---|---|---|---|---|---|---|---|---|---|---|---|
| | OpenAI | Claude | Mistral | OpenAI | Claude | Google | LLaMA3 | Qwen2.5 | Yi1.5 | | |
| **Oracle** | 63.17 | 52.98 | 34.69 | 64.30 | 63.75 | 66.74 | 64.11 | 83.32 | 64.12 | 91.91 | 64.91 |
| **Random** | 36.47 | 34.56 | 27.76 | 48.94 | 41.89 | 45.15 | 37.86 | 41.98 | 33.94 | 54.93 | 40.35 |
| **kNN (k=10)** | 57.33 | 52.82 | 34.27 | 53.93 | 52.66 | 50.92 | 41.45 | 39.67 | 34.98 | 74.22 | 49.23 |
| **kNN (k=100)** | 57.38 | 52.77 | **34.26** | 54.73 | 53.40 | 52.18 | 48.98 | 56.18 | 39.74 | 77.22 | 52.68 |
| **Linear** | **57.60** | **52.84** | 34.26 | **55.61** | **53.54** | **52.64** | 48.94 | **56.46** | 41.83 | **77.68** | **53.14** |
| **Linear (MF)** | 56.85 | 52.08 | 33.76 | 55.31 | 53.40 | 51.93 | 48.94 | 55.91 | **41.88** | 77.27 | 52.73 |
| **MLP** | 55.50 | 51.29 | 32.18 | 54.43 | 50.82 | 51.51 | 48.52 | 54.62 | 41.19 | 77.08 | 51.71 |
| **MLP (MF)** | 57.50 | 52.84 | 34.20 | 55.60 | 53.21 | 51.46 | 48.71 | 56.21 | 41.63 | 77.40 | 52.88 |
| **Graph (k=10)** | 55.66 | 45.64 | 31.50 | 53.34 | 45.19 | 50.45 | 40.06 | 54.99 | 33.62 | 76.24 | 48.67 |
| **Graph (k=100)** | 57.37 | 51.84 | 32.25 | 52.70 | 51.89 | 51.41 | **49.10** | 55.24 | 39.52 | 76.87 | 51.82 |
| **Attn (k=10)** | 54.43 | 45.12 | 26.52 | 53.07 | 52.35 | 51.89 | 41.09 | 48.00 | 34.58 | 73.53 | 48.06 |
| **Attn (k=100)** | 55.29 | 52.49 | 27.94 | 52.23 | 52.12 | 50.61 | 45.12 | 56.03 | 32.65 | 77.37 | 50.18 |
| **D-Attn (k=10)** | 54.50 | 44.62 | 29.34 | 53.31 | 50.17 | 51.38 | 39.54 | 32.05 | 34.81 | 74.40 | 46.41 |
| **D-Attn (k=100)** | 57.17 | 52.77 | 24.45 | 51.49 | 49.43 | 51.82 | 49.04 | 23.90 | 34.92 | 77.48 | 47.25 |

**Neural Network Methods**   To explore non-linear routing strategies, we implement neural architectures that can capture complex patterns. The **MLP Router** uses a feed-forward network with multiple hidden layers (100 dimensions each) to learn non-linear mappings between query embeddings and routing outcomes, balancing expressiveness with computational efficiency. The **MLP Matrix Factorization (MF) Router** extends the linear MF approach by processing prompt-model embedding interactions through multi-layer perceptrons, enabling more sophisticated modeling of query-model relationships while maintaining interpretable factorized structure.

**Graph-Based and Attention Methods**   For the most sophisticated approaches, we evaluate methods that explicitly model relationships between queries and models. The **Graph Router** (Feng et al., 2024) represents routing as a bipartite graph where queries and models form nodes with utility-representing edges, using graph neural networks to capture complex patterns between similar queries and model performance.

We also implement two attention-based architectures leveraging support sets. The **Attentive Router** uses permutation-invariant attention modules to process support examples (prompt-utility pairs) with cross-attention between target prompts and support examples for contextual routing decisions. The **Double Attentive Router** extends this with dual attention mechanisms across both queries and candidate models, capturing similarities between queries and relationships between different models' performance characteristics on related inputs.

**Implementation**   All approaches use consistent query representations: BERT embeddings for text-only routing and VLM2Vec for multi-modal scenarios, isolating algorithmic impact while ensuring fair comparison. Detailed specifications, hyperparameter settings, and training protocols are in Appendix C.

## 6 QUANTITATIVE RESULTS

In this section, we present a comprehensive evaluation of diverse routing approaches using utility prediction formulation, which provides the most reliable assessment of routing performance by revealing complete cost-performance trade-offs. For textual queries, we use BERT embeddings (Devlin et al., 2019) to represent the input text, while for multi-modal queries, we employ VLM2Vec (Jiang et al., 2024) to extract unified representations of image-text pairs. For completeness, selection-based routing results across all benchmarks are provided in Appendix D, though we emphasize that utility prediction provides more reliable routing assessment.

**Text-Based Routing Results**   Tables 2 presents the performance of various routing approaches on our text-based benchmarks using AUC of the Pareto front, which comprehensively captures each router's ability to balance performance and cost across the entire spectrum of preference settings.

The most striking finding is the remarkable effectiveness of simple kNN-based routing. kNN with k=100 achieves an average AUC score of 52.68 across all benchmarks, which is competitive with or superior to more complex approaches like MLP (51.71) and Graph Neural Networks (51.82). Linear models also demonstrate surprisingly strong performance, achieving an average AUC of 53.14.

The relative performance ranking of routing methods remains consistent across different benchmarks and model pools. When kNN outperforms more complex methods on one benchmark, it typically maintains this advantage on others, suggesting that the effectiveness of simple routing approaches is robust across different evaluation settings.

Support set size has a consistent positive impact. Increasing k from 10 to 100 generally improves performance for kNN-based routing, demonstrating the value of leveraging larger neighborhoods for routing decisions.

**Computational Efficiency Analysis** Beyond routing accuracy, computational efficiency is crucial for practical deployment. Table 3 reports the cumulative inference time required to route all examples in Routerbench. We measured routing time only, excluding one-time costs of building indices and training models. For fair comparison, CPU-based methods (kNN, Linear, MLP) were run on Intel Xeon Platinum 8275CL processors, while GPU-accelerated methods (Graph, attention-based) used NVIDIA A100 GPUs.

Table 3: The cumulative time to route all examples in Routerbench.

| | Latency(s) |
|---|---|
| **kNN (k=10)** | 75.76 |
| **kNN (k=100)** | 65.69 |
| **Linear** | 84.51 |
| **Linear (MF)** | 95.71 |
| **MLP** | 116.44 |
| **MLP (MF)** | 92.52 |
| **Graph (k=10)** | 866.34 |
| **Graph (k=100)** | 872.03 |
| **Attn (k=10)** | 877.64 |
| **Attn (k=100)** | 883.31 |
| **D-Attn (k=10)** | 905.51 |
| **D-Attn (k=100)** | 906.32 |

The efficiency differences are dramatic. kNN (k=100) requires only 65.69 seconds, making it the fastest method overall. Simple parametric models require 84-95 seconds, while graph and attention-based methods require over 866 seconds—approximately 13-14× slower than kNN.

For deployment scenarios requiring thousands of routing decisions per second, these latency differences become critical. The combination of competitive routing performance and superior computational efficiency makes simple methods particularly attractive for high-throughput applications.

**Robustness Under Distribution Shift** To evaluate generalization under distribution shift, we conducted cross-dataset evaluation where models trained on one Router-Bench dataset are tested on others. This creates 36 train-test pairs per routing method, allowing us to measure performance degradation when moving from in-distribution to out-of-distribution queries.

Table 4: Average AUC for in-distribution (ID) and out-of-distribution (OOD) test sets.

| Model | ID | OOD | Δ |
|---|---|---|---|
| **kNN (k=10)** | 73.53 | 70.41 | 3.13 |
| **kNN (k=100)** | 76.54 | 73.91 | 2.63 |
| **Linear** | 77.03 | 73.70 | 3.33 |
| **Linear (MF)** | 76.63 | 69.96 | 6.67 |
| **MLP** | 76.45 | 72.23 | 4.22 |
| **MLP (MF)** | 76.77 | 71.47 | 5.30 |
| **Graph (k=10)** | 75.58 | 70.39 | 5.19 |
| **Graph (k=100)** | 76.20 | 72.38 | 3.82 |
| **Attn (k=10)** | 72.87 | 67.93 | 4.94 |
| **Attn (k=100)** | 76.71 | 72.19 | 4.52 |
| **D-Attn (k=10)** | 73.74 | 68.58 | 5.16 |
| **D-Attn (k=100)** | 76.81 | 72.32 | 4.49 |

Table 4 shows that all routing methods experience performance drops under distribution shift, but the magnitude varies significantly. kNN (k=100) shows the smallest degradation (2.63 points), while Linear Matrix Factorization suffers the largest drop (6.67 points). Complex methods like attention-based routers show intermediate degradation (4.49-4.94 points).

This pattern suggests that non-parametric methods may be inherently more robust to domain shifts. Unlike parametric models that learn fixed global patterns, kNN can adapt locally to new query distributions by leveraging the most relevant examples from the training set.

**Multi-Modal Routing Results** Tables 5 shows performance on our vision-language model routing benchmarks using utility prediction evaluation. Simple methods maintain their effectiveness in multi-modal settings, with kNN (k=100) achieving an average AUC of 72.12—outperforming most neural approaches.

Multi-modal routing confirms the patterns observed in text-only scenarios. The effectiveness of simple methods extends seamlessly to complex multi-modal inputs, where optimal model selection depends on both textual content and visual characteristics.

Attention-based architectures show particular strength on certain multi-modal tasks like MME, likely due to their ability to model interactions between visual and textual components more explicitly than simpler approaches.

Table 5: AUC scores for vision language benchmark using utility prediction routing. Higher is better.

| | Blink | | Flickr30k | | MathVista | | MME | | MMMU | | Avg |
|---|---|---|---|---|---|---|---|---|---|---|---|
| | OpenAI | Claude | OpenAI | Claude | OpenAI | Claude | OpenAI | Claude | OpenAI | Claude | |
| **Oracle** | 98.53 | 92.97 | 78.00 | 73.91 | 89.30 | 63.66 | 97.79 | 90.17 | 92.30 | 80.28 | 85.69 |
| **Random** | 71.82 | 57.72 | 54.12 | 49.50 | 52.91 | 36.96 | 75.21 | 52.05 | 64.47 | 47.76 | 56.25 |
| **kNN (k=10)** | 83.91 | 77.47 | 59.29 | 61.41 | 65.88 | 49.57 | 90.84 | 84.52 | 75.04 | 58.26 | 70.62 |
| **kNN (k=100)** | 84.79 | 78.34 | 58.89 | 61.16 | **72.96** | 50.29 | 90.84 | 84.56 | 78.11 | 61.27 | **72.12** |
| **Linear** | **85.48** | 77.05 | 59.42 | **61.69** | 70.85 | 50.28 | 91.29 | 85.19 | 75.05 | 60.54 | 71.68 |
| **Linear (MF)** | 84.62 | 76.76 | 58.82 | 61.30 | 64.46 | 48.20 | **92.19** | 84.51 | 71.61 | 60.16 | 70.26 |
| **MLP** | 78.86 | 73.45 | 59.50 | 58.87 | 60.20 | 48.89 | 91.96 | 83.27 | 68.12 | 57.95 | 68.11 |
| **MLP (MF)** | 84.96 | 77.72 | 60.29 | 61.61 | 64.45 | 49.61 | 91.29 | **85.62** | 74.66 | 61.67 | 71.19 |
| **Graph (k=10)** | 83.55 | 76.45 | 58.66 | 58.34 | 64.48 | 44.71 | 90.79 | 84.02 | 75.04 | 62.04 | 69.81 |
| **Graph (k=100)** | 84.79 | **78.65** | 58.56 | 61.40 | 70.85 | 48.22 | 91.06 | 84.56 | 77.73 | 61.67 | 71.75 |
| **Attn (k=10)** | 84.08 | 77.14 | 58.31 | 58.29 | 66.57 | 46.10 | 91.06 | 84.75 | 73.87 | 58.26 | 69.84 |
| **Attn (k=100)** | 84.95 | 78.16 | 58.59 | 54.42 | 71.54 | 50.97 | 90.84 | 84.59 | **79.26** | 61.29 | 71.46 |
| **D-Attn (k=10)** | 83.56 | 76.99 | 59.90 | 59.37 | 67.27 | 48.87 | 90.82 | 84.75 | 73.80 | 58.28 | 70.36 |
| **D-Attn (k=100)** | 84.79 | 78.32 | 58.72 | 51.93 | 70.12 | **50.97** | 89.87 | 84.59 | 73.41 | 61.28 | 70.40 |

**Statistical Validation and Embedding Analysis**  To ensure our findings are robust, we conducted statistical validation across five independent runs. The results confirm consistent relative performance: kNN (k=100) achieves 77.31 ± 0.27 AUC, Linear achieves 77.52 ± 0.21 AUC, and MLP achieves 76.94 ± 0.33 AUC. The small standard deviations indicate that our conclusions are not dependent on particular random seeds or data splits.

Our analysis shows that switching from BERT to SFR embeddings provides modest improvements, but importantly, the relative ranking of routing methods remains consistent across embedding types (detailed results in Appendix H).

## 7 THEORETICAL ANALYSIS

In this section, we develop a theoretical framework to explain why simple kNN-based routers often match or outperform more complex learned routers. Our analysis addresses a fundamental question: under what conditions does the local structure of the query-performance space provide sufficient signal for effective routing?

We begin by formalizing the locality property that underlies effective kNN routing:

**Definition 1** ($\delta$-Locality). *Given a query embedding space $\mathcal{X}$, utility function $u(x, m)$, and distance function $d(\cdot, \cdot)$, model performance exhibits $\delta$-locality if for any two queries $x_1$ and $x_2$:*

$$d(x_1, x_2) < \delta \implies |u(x_1, m) - u(x_2, m)| < \epsilon(\delta)$$

*where $\epsilon(\delta)$ is a monotonically increasing function with $\epsilon(0) = 0$.*

This definition captures the intuition that semantically similar queries should yield similar utility scores from the same model. If this property holds, then kNN methods can make effective routing decisions by leveraging the performance patterns of nearby queries.

We now compare the sample complexity of kNN routers to parametric alternatives:

**Theorem 1** (Sample Complexity). *For a query distribution $\mathcal{D}$ with $\delta$-locality in utility space:*

*(a) A kNN router requires a training sample size of $\Theta\left(\frac{C_{\mathcal{X},d}}{\delta^d} \cdot \log\left(\frac{1}{\alpha}\right)\right)$ to achieve expected regret $O(\epsilon(\delta))$ with probability $1 - \alpha$, where $d$ is the intrinsic dimension of the embedding space and $C_{\mathcal{X},d}$ is a constant depending on the space.*

*(b) A parametric router with $L$ Lipschitz-continuous layers requires a training sample size of $\Omega(L/\epsilon(\delta)^2)$ to achieve the same regret bound.*

The key insight is that when the embedding space has low intrinsic dimension $d$ and strong locality properties (where $\epsilon(\delta)$ decreases rapidly with $\delta$), kNN routers require significantly fewer training samples than parametric routers to achieve the same performance guarantees. The full proof is provided in Appendix I.

**Empirical Validation**  We provide empirical evidence supporting our theoretical framework. Figure 1 demonstrates that as embedding distance between query pairs increases, the agreement between their model performance rankings decreases. We measure agreement as the correlation between performance scores across all models for each query pair, then bin query pairs by embedding distance.

The strong negative correlation (r=-0.815 for Arc-Challenge, r=-0.875 for GSM) provides clear empirical support for the $\delta$-locality property: queries close in embedding space tend to have similar relative model performance patterns.

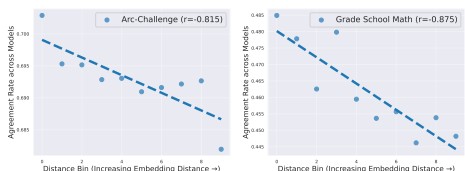

Figure 1: As embedding distance between query pairs increases, the agreement between their model performance rankings decreases, demonstrating locality in the query-performance space.

To validate the intrinsic dimensionality assumption in our theorem, we analyzed our embedding spaces using the TwoNN method (Facco et al., 2017). RouterBench embeddings have intrinsic dimensions of approximately 2-28, while vision-language embeddings have intrinsic dimensions of 13-18—substantially lower than their ambient dimensions (768 for BERT, 3584 for VLM2Vec).

These empirical findings directly support our theoretical predictions. The observed low intrinsic dimensions ($d \approx 2 - 28$) combined with strong locality properties create the favorable conditions where kNN methods achieve sample complexity advantages over parametric approaches, explaining our empirical results showing competitive performance with significantly fewer parameters.

Our theoretical analysis explains why simple methods are effective: locality properties in embedding spaces make neighborhood-based approaches highly suitable for routing, particularly when combined with low intrinsic dimensionality and limited training data.

## 8  DISCUSSION AND CONCLUSION

This paper fundamentally rethinks LLM routing by challenging the assumption that sophisticated architectures are necessary for effective model selection. Through comprehensive evaluation across text-only and multi-modal benchmarks, we demonstrate that simple kNN-based routers consistently match or outperform complex learned approaches while offering substantial practical advantages.

**Key Findings**  Our investigation reveals that kNN routing achieves competitive performance (52.68 AUC on text, 72.12 on multi-modal benchmarks) while being 13-14× faster than complex alternatives. Most importantly, simple methods demonstrate superior robustness under distribution shift, with kNN showing only 2.63-point degradation compared to 6.67 points for complex methods.

Our analysis explains these findings: the strong locality properties in embedding spaces—where semantically similar queries benefit from similar models—enable kNN methods to achieve effective routing with lower sample complexity. The combination of low intrinsic dimensionality ($d \approx 2-28$) and strong locality creates favorable conditions for neighborhood-based routing.

**Limitations**  While our results favor simple methods, complex approaches may be justified in specific scenarios: long-tail queries with sparse semantic neighborhoods, rapidly changing performance landscapes, or tasks requiring compositional reasoning across multiple domains. Additionally, kNN faces scalability constraints where memory requirements grow linearly with support set size.

**Implications**  Our findings challenge the field's trend toward architectural complexity, suggesting that research should focus on understanding fundamental routing properties rather than developing sophisticated architectures. The effectiveness of kNN routing reinforces a key principle: complex solutions should be justified by meaningful improvements over simple alternatives. For practitioners, the combination of competitive performance, superior efficiency, and enhanced robustness makes simple methods particularly attractive for deployment, potentially democratizing access to effective routing for organizations with limited resources. Future work should prioritize improving embedding quality, investigating alternative training signals, and developing methods to identify when complexity is truly necessary.

ETHICS STATEMENT

This research exclusively utilizes publicly available datasets and benchmarks in full compliance with their respective licenses and terms of use. No personally identifiable information, sensitive data, or proprietary datasets were collected, generated, or analyzed. All experimental procedures adhere to standard academic research practices and pose no ethical concerns regarding data privacy, misuse, or harm.

REPRODUCIBILITY STATEMENT

We have made extensive efforts to ensure the reproducibility of our work. All experimental details, including hyperparameter settings, training protocols, and model architectures, are provided in Appendix C. Complete benchmark construction methodologies, model selection criteria, and evaluation protocols are detailed in Appendix B. The theoretical analysis includes formal definitions, theorem statements, and complete proofs in Appendix I. All datasets used are publicly available, with specific data splits, preprocessing steps, and evaluation metrics documented in the appendices. Statistical validation procedures, including multiple runs and standard deviation calculations, are described in Section 6. All source code, model implementations, and evaluation scripts are provided in the supplementary materials to enable immediate replication of our results. Random seeds are fixed and documented throughout all experiments to ensure deterministic outcomes.

USE OF LLMS

Large language models were employed as writing assistance tools to enhance the clarity, coherence, and presentation quality of this manuscript. All core technical contributions, experimental design, methodology, analysis, and conclusions represent original work conducted entirely by the authors. LLM assistance was strictly limited to grammar correction, style refinement, and clarity improvements, without modification of technical content, experimental results, or research conclusions. The intellectual contributions and scientific validity of this work remain solely attributable to the authors.

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

## A  ADDITIONAL RELATED WORKS

Beyond the core routing approaches discussed in Section 2, several other research directions are relevant to our investigation of LLM routing mechanisms.

**LLM Inference Optimization**   Routing can be viewed as one component of the broader challenge of optimizing LLM inference. Complementary approaches include speculative decoding (Leviathan et al., 2023), quantization (Egashira et al., 2024), and hardware-specific optimizations. Our work on efficient routing complements these techniques, as the benefits of selecting the most appropriate model can be combined with optimizations to the inference process itself.

**Connections to Recommendation Systems**   LLM routing shares fundamental similarities with recommendation systems, where the goal is to match users (queries) with items (models) that maximize utility. Several techniques from recommender systems research have direct analogs in routing approaches. Matrix factorization methods like those used in RouteLLM (Ong et al., 2024) mirror collaborative filtering techniques in recommendation (Su & Khoshgoftaar, 2009). Similarly, our kNN approach resembles item-based neighborhood methods that recommend items based on similarity to previously rated items (Sarwar et al., 2001). Content-based recommendation techniques that leverage item features parallel our embedding-based routing approaches. Even the dual optimization of performance and cost in routing mirrors multi-objective recommendation systems that balance relevance with diversity or novelty (De Myttenaere et al., 2014). This connection offers promising opportunities to adapt proven recommendation techniques to the routing domain, particularly for handling cold-start problems with new queries or models, and for developing effective hybrid routing strategies.

## B  ROUTING BENCHMARKS

In this section, we provide detailed information about our benchmark construction methodology, model selection criteria, and evaluation protocols for both text and vision-language routing benchmarks.

### B.1  TEXT-BASED ROUTING BENCHMARKS

We construct comprehensive benchmarks based on three established LLM evaluation frameworks: AlpacaEval, Open LLM Leaderboard v2, and HELM-Lite, as well as incorporating RouterBench. Our goal is to provide a diverse set of routing scenarios that reflect real-world deployment challenges.

**Model Selection**   For each benchmark, we select three model families with multiple variants per family to represent realistic routing scenarios:

- **AlpacaEval:** We include OpenAI models, Claude models, and Mistral models.
- **Open LLM Leaderboard:** We include LLaMA3 variants, Qwen2.5 variants, and Yi-1.5 variants.
- **HELM-Lite:** We include OpenAI models, Claude models, and Google Gemini models.
- **RouterBench:** We use all 11 models provided in the original benchmark.

Please refer to Table B.1 for the list of models and their costs.

**Task Coverage**   Our benchmarks span a wide range of task categories:

- **AlpacaEval:** Instruction following tasks evaluated through human preference alignment.
- **Open LLM Leaderboard:** Mathematical reasoning, knowledge-based reasoning and instruction following tasks.
- **HELM-Lite:** Knowledge-intensive tasks, reasoning tasks, and instruction following capabilities.
- **RouterBench:** Six tasks spanning mathematical reasoning, code generation, knowledge, and commonsense reasoning.

Table B.1: Candidate models and their costs for text-based routing benchmark.

| Benchmark | Model Family | Candidate Models | Input cost ($ per 1M tokens) | Output cost ($ per 1M tokens) |
|---|---|---|---|---|
| ALpacaEval | OpenAI Family | gpt-3.5-turbo-0301 | 1.5 | 2.0 |
| | | gpt-3.5-turbo-0613 | 1.5 | 2.0 |
| | | gpt-3.5-turbo-1106 | 1.0 | 2.0 |
| | | gpt-4-0125-preview | 10 | 30 |
| | | gpt-4o-2024-05-13 | 5 | 15 |
| | | gpt-4 | 30 | 60 |
| | | gpt-4-0314 | 30 | 60 |
| | | gpt-4-0613 | 30 | 60 |
| | | gpt-4-1106-preview | 10 | 30 |
| | Claude Family | claude-2 | 8 | 24 |
| | | claude-2.1 | 8 | 24 |
| | | claude-3-5-sonnet-20240620 | 3 | 15 |
| | | claude-3-opus-20240229 | 15 | 75 |
| | | claude-3-sonnet-20240229 | 3 | 15 |
| | | claude-instant-1.2 | 0.8 | 2.4 |
| | Mistral Family | Mistral-7B-Instruct-v0.2 | 0.25 | 0.25 |
| | | Mixtral-8x22B-Instruct-v0.1 | 2 | 6 |
| | | Mixtral-8x7B-Instruct-v0.1 | 0.7 | 0.7 |
| | | mistral-large-2402 | 8 | 24 |
| | | mistral-medium | 2.7 | 8.1 |
| Open LLM Leaderboard v2 | Qwen2.5 | Qwen2.5-0.5B-Instruct | 0.08 | 0.08 |
| | | Qwen2.5-1.5B-Instruct | 0.2 | 0.2 |
| | | Qwen2.5-7B-Instruct | 0.3 | 0.3 |
| | | Qwen2.5-14B-Instruct | 0.8 | 0.8 |
| | | Qwen2.5-32B-Instruct | 0.8 | 0.8 |
| | | Qwen2.5-72B-Instruct | 1.2 | 1.2 |
| | Llama3 | Llama-3-8B-Instruct | 0.2 | 0.2 |
| | | Llama-3-70B-Instruct | 0.9 | 0.9 |
| | Yi1.5 | Yi-1.5-6B-Chat | 0.3 | 0.3 |
| | | Yi-1.5-9B-Chat | 0.4 | 0.4 |
| | | Yi-1.5-34B-Chat | 0.8 | 0.8 |
| HELM-Lite | OpenAI Family | gpt-4o-2024-05-13 | 5.0 | 15.0 |
| | | gpt-4o-mini-2024-07-18 | 0.15 | 0.6 |
| | | gpt-3.5-turbo-0613 | 1.5 | 2.0 |
| | | gpt-4-0613 | 30 | 60 |
| | | gpt-4-turbo-2024-04-09 | 10 | 30 |
| | | gpt-4-1106-preview | 10 | 30 |
| | Claude Family | claude-3-5-sonnet-20240620 | 3 | 15 |
| | | claude-3-opus-20240229 | 15 | 75 |
| | | claude-3-sonnet-20240229 | 3 | 15 |
| | | claude-3-haiku-20240307 | 0.25 | 1.25 |
| | | claude-2 | 8 | 24 |
| | | claude-instant-v1 | 0.8 | 2.4 |
| | | claude-v1.3 | 8 | 24 |
| | | claude-2.1 | 8 | 24 |
| | | claude-instant-1.2 | 0.8 | 2.4 |
| | Google Family | gemini-1.0-pro-002 | 0.5 | 1.5 |
| | | gemini-1.0-pro-001 | 0.5 | 1.5 |
| | | gemini-1.5-pro-001 | 3.5 | 10.5 |
| | | gemini-1.5-flash-001 | 0.075 | 0.3 |
| | | text-bison-001 | 0.5 | 1.5 |
| | | text-unicorn-001 | 7.0 | 21.0 |
| | | gemma-2-9b-it | 0.2 | 0.2 |
| | | gemma-2-27b-it | 0.6 | 0.6 |
| | | gemma-7b | 0.1 | 0.1 |
| RouterBench | RouterBench | gpt-3.5 | 1.0 | 2.0 |
| | | claude-instant-v1 | 0.8 | 2.4 |
| | | claude-v1 | 8.0 | 24.0 |
| | | claude-v2 | 8.0 | 24.0 |
| | | gpt-4 | 10.0 | 30.0 |
| | | llama-70b | 0.9 | 0.9 |
| | | Mixtral-8x7B | 0.6 | 0.6 |
| | | Yi-34B | 0.8 | 0.8 |
| | | WizardLM-13B | 0.3 | 0.3 |
| | | code-llama-34B | 0.776 | 0.776 |
| | | Mistral-7B | 0.2 | 0.2 |

**Performance Scoring Methodology** For each benchmark, we use evaluation metrics aligned with their respective leaderboards:

- **AlpacaEval:** We use length-controlled win rates against the reference model.
- **Open LLM Leaderboard:** We use accuracy metrics for each task as reported in the leaderboard.
- **HELM-Lite:** We use the benchmark-specific metrics as reported in the leaderboard.
- **RouterBench:** We use the performance scores provided in the original benchmark.

**Cost Calculation**  We calculate costs based on actual pricing of each API, using:

$$c(x, m) = \text{InputTokens} \times \text{InputPrice}_m + \text{OutputTokens} \times \text{OutputPrice}_m$$

For open-source models, we estimate costs using the pricing of equivalent commercial offerings from TogetherAI.

### B.2  VISION-LANGUAGE ROUTING BENCHMARKS

To address the growing importance of multi-modal systems, we develop the first benchmark for routing between vision-language models. This benchmark assesses how routing methods perform when inputs span different modalities. We utilize the existing evaluation outcomes from vHELM leaderboard.

**Model Selection**  We focus on two leading vision-language model families - OpenAI models and Claude models - with multiple variants representing different capability-cost tradeoffs. Table B.2 list the candidate models and their costs.

Table B.2: Candidate models and their costs for VLM routing benchmark.

| Benchmark | Model Family | Candidate models | Input cost ($ per 1M tokens) | Output cost ($ per 1M tokens) |
|---|---|---|---|---|
| vHELM | OpenAI Family | gpt-4-turbo-2024-04-09 | 10 | 30 |
| | | gpt-4.1-2025-04-14 | 2 | 8 |
| | | gpt-4.1-mini-2025-04-14 | 0.4 | 1.6 |
| | | gpt-4.1-nano-2025-04-14 | 0.1 | 0.4 |
| | | gpt-4.5-preview-2025-02-27 | 75 | 150 |
| | | gpt-4o-2024-05-13 | 5 | 15 |
| | | gpt-4o-2024-08-06 | 2.5 | 10 |
| | | gpt-4o-2024-11-20 | 2.5 | 10 |
| | | gpt-4o-mini-2024-07-18 | 0.15 | 0.6 |
| | | o1-2024-12-17 | 15 | 60 |
| | | o3-2025-04-16 | 10 | 40 |
| | | o4-mini-2025-04-16 | 1.1 | 4.4 |
| | Claude Family | claude-3-5-sonnet-20240620 | 3 | 15 |
| | | claude-3-5-sonnet-20241022 | 3 | 15 |
| | | claude-3-7-sonnet-20250219 | 3 | 15 |
| | | claude-3-7-sonnet-20250219-thinking-64k | 3 | 15 |
| | | claude-3-haiku-20240307 | 0.8 | 4 |
| | | claude-3-opus-20240229 | 15 | 75 |
| | | claude-3-sonnet-20240229 | 3 | 15 |

**Dataset Selection**  We select five diverse vision-language datasets with varying task complexity:

- **Blink:** Tests basic visual perception and object recognition capabilities.
- **Flickr30k:** Evaluates natural image description and scene understanding.
- **MathVista:** Challenges models with mathematical reasoning over visual inputs.
- **MME:** A comprehensive evaluation benchmark covering multiple vision-language capabilities.
- **MMMU:** Tests multi-modal understanding across challenging academic domains.

Each dataset presents unique routing challenges, as models show different strengths across visual understanding tasks. For example, some models excel at detailed image description but struggle with visual reasoning tasks.

### B.3  EVALUATION PROTOCOL

#### B.3.1  AUC SCORE CALCULATION METHODOLOGY

To evaluate utility prediction approaches, we calculate the area under the curve (AUC) of the non-decreasing convex hull in the cost-performance space using the following procedure:

1. For a given query $x$, we obtain predicted utility scores $\hat{u}(x, m) = \hat{s}(x, m) - \lambda \times \hat{c}(x, m)$ for each model $m \in \mathcal{M}$ across various values of $\lambda$.

2. For each $\lambda$ value, we select the model with the highest predicted utility: $m_\lambda = \arg\max_{m \in \mathcal{M}} \hat{u}(x, m)$.

3. We plot the actual performance-cost pairs $(c(x, m_\lambda), s(x, m_\lambda))$ in the cost-performance space.

4. We compute the non-decreasing convex hull of these points to obtain the Pareto-optimal frontier.

5. The AUC is calculated as the area under this frontier, normalized so that the maximum score is 100 and the maximum cost is 1.

This approach ensures that routers are evaluated on their ability to make optimal trade-offs across the entire spectrum of cost-performance preferences.

### B.4 DATA SPLITS AND REPRODUCIBILITY

To ensure reproducible evaluation, we use the following data splits:

- For all benchmarks, we create random splits over prompts with 70% training, 10% validation, and 20% test data.
- For support set experiments, we ensure no leakage between support sets and test queries.
- All random seeds are fixed and documented in our code to ensure reproducibility.

Our benchmark construction methodology ensures comprehensive evaluation across diverse tasks, models, and modalities, providing a robust foundation for comparing different routing approaches.

## C ROUTING APPROACHES

This section provides detailed implementation information for all routing approaches evaluated in our study, including architecture specifications, hyperparameter settings, training procedures, and computational requirements.

### C.1 INPUT REPRESENTATIONS

For all routing approaches, we use consistent input representations to ensure fair comparison:

**Text Embeddings** For text-only queries, we primarily use BERT (Devlin et al., 2019) base model (768-dimensional embeddings) to encode input queries. We take the [CLS] token embedding as the query representation. In our ablation studies (Table H.1), we also experiment with SFR (Meng et al., 2024) embeddings (4096-dimensional) to assess the impact of embedding quality on routing performance. All embeddings are L2-normalized to unit length.

**Vision-Language Embeddings** For multi-modal queries, we employ VLM2Vec (Jiang et al., 2024) to extract unified representations of image-text pairs. Specifically, we utilize the Qwen7B-based variant, which generates 3584-dimensional embeddings that effectively capture both textual content and visual features in a shared embedding space.

### C.2 ROUTING MODEL ARCHITECTURES

**kNN Router** Our kNN implementation uses cosine similarity to identify the nearest neighbors in the embedding space. For utility prediction, we compute the weighted average of performance scores and costs from the $k$ nearest neighbors:

$$\hat{s}(x, m) = \frac{1}{k} \sum_{i=1}^{k} s(x_i, m), \quad \hat{c}(x, m) = \frac{1}{k} \sum_{i=1}^{k} c(x_i, m)$$

For model selection, we use the majority voting mechanism, where each neighbor votes for the model that maximizes its utility score.

**Linear Router** For utility prediction, we implement a linear regression model that maps query embeddings directly to performance scores and costs:

$$\hat{s}(x, m) = W_s^m \cdot \text{emb}(x) + b_s^m, \quad \hat{c}(x, m) = W_c^m \cdot \text{emb}(x) + b_c^m \quad \text{(C.1)}$$

where $W_s^m$, $W_c^m \in \mathbb{R}^{768}$ and $b_s^m$, $b_c^m \in \mathbb{R}$ are learnable parameters for each model $m$. For model selection, we use a multi-class logistic regression:

$$p(m|x) = \text{softmax}(W \cdot \text{emb}(x) + b)$$

where $W \in \mathbb{R}^{M \times 768}$ and $b \in \mathbb{R}^M$ are learnable parameters.

**Linear Matrix Factorization (MF) Router** This approach assigns a learnable embedding to each model and predicts performance through the interaction between query and model embeddings:

$$\hat{s}(x, m) = \text{emb}(x)^\top \cdot W_s \cdot \text{emb}(m) + b_s, \quad \hat{c}(x, m) = \text{emb}(x)^\top \cdot W_c \cdot \text{emb}(m) + b_c$$

where $W_s, W_c \in \mathbb{R}^{768 \times d_m}$, $\text{emb}(m) \in \mathbb{R}^{d_m}$ is the learnable embedding for model $m$ (we use $d_m = 128$), and $b_s, b_c \in \mathbb{R}$ are learnable biases.

**MLP Router** Our MLP consists of 3 fully-connected layers with ReLU activations:

$$\hat{s}(x, m) = \text{MLP}_s^m(\text{emb}(x)), \quad \hat{c}(x, m) = \text{MLP}_c^m(\text{emb}(x))$$

The architecture uses 100 dimensions for each hidden layer.

**MLP Matrix Factorization (MF) Router** Similar to Linear MF, but replaces the linear projection with an MLP:

$$\hat{s}(x, m) = \text{MLP}_s([\text{emb}(x); \text{emb}(m)]), \quad \hat{c}(x, m) = \text{MLP}_c([\text{emb}(x); \text{emb}(m)])$$

where $[;]$ denotes concatenation. The MLP has architecture use 100 hidden units with ReLU activations.

**Graph Router** We implement a bipartite graph neural network where queries and models form nodes. Each query-model pair is connected by directed edges containing features that represent scores, token counts, and neighborhood information. Our implementation uses the GeneralConv class from PyTorch Geometric with the following architecture:

$$h_x^{(0)} = W_{\text{proj}} \cdot \text{emb}(x), \quad h_m^{(0)} = \text{emb}(m)$$
$$e_{xm}^{(0)} = W_{\text{edge}} \cdot [s_{xm}, t_{xm}, \text{mask}_{xm}]$$
$$h_i^{(l+1)} = \sigma \left( \text{BN}^{(l)} \left( \text{GNN}^{(l)}(h_i^{(l)}, h_j^{(l)} | j \in \mathcal{N}(i), e_{ij}^{(l)}) \right) \right)$$

where $\text{emb}(m)$ is a learned embedding for each model, $s_{xm}$ and $t_{xm}$ are the score and token count features between query $x$ and model $m$, $\text{mask}_{xm}$ indicates whether the edge features are observed or missing, BN is batch normalization, and $\sigma$ is ReLU. The graph is enriched with $k$-nearest neighbor information from a pre-computed benchmark dataset. After message passing through multiple GNN layers (we use 2 layers with hidden dimension 128), the final prediction is computed using an MLP over the concatenated node representations of each query-model pair.

**Attentive Router** This approach employs a Conditional Neural Process architecture with both self-attention and cross-attention mechanisms to process nearest neighbor examples:

$$\text{support} = (x_i, s(x_i, m), c(x_i, m)) | x_i \in \text{kNN}(x, k)$$
$$Z = \text{SelfAttention}(\text{support})$$
$$\text{latent}(x, m) = \text{CrossAttention}(Q = \text{emb}(x), K = \text{emb}(\text{support}), V = Z)$$
$$\hat{s}(x, m) = \text{MLP}_s(\text{latent}(x, m)), \quad \hat{c}(x, m) = \text{MLP}_c(\text{latent}(x, m)),$$

where the self-attention module applies permutation-invariant processing to the support set, and the cross-attention module allows the query to attend to the processed support set. For each model, we retrieve $k$-nearest neighbors with their corresponding scores and token counts, and project them to a shared embedding space. The architecture uses multi-head attention (4 heads with dimension 32 per head), followed by MLP prediction heads for both score and token count estimation.

**Double Attentive Router**    Extends the Attentive Router by processing the support set with a dual attention mechanism that captures both query-level and model-level interactions:

$$\text{support} = (x_i, s(x_i, m), c(x_i, m)) | x_i \in \text{kNN}(x, k), m \in \mathcal{M}$$
$$Z = \text{DoubleAttention}(\text{support})$$
$$\text{latent}(x, m) = \text{CrossAttention}(Q = \text{emb}(x), K = \text{emb}(\text{support}), V = Z)$$
$$\hat{s}(x, m) = \text{MLP}_s(\text{latent}(x, m)), \quad \hat{c}(x, m) = \text{MLP}_c(\text{latent}(x, m))$$

where the double attention mechanism applies attention operations across both examples and models, allowing for richer representations that capture cross-model dependencies. The support set is organized as a 3D tensor (batch × models × examples) and processed with sequential attention operations. This architecture enables the router to model complex interactions between different models on similar queries, thereby improving routing accuracy across the model pool.

### C.3   TRAINING PROTOCOLS

**Loss Functions**    For utility prediction models, we use mean squared error (MSE) loss for both score and cost prediction:

$$\mathcal{L} = \text{MSE}(\hat{s}(x, m), s(x, m)) + \alpha \cdot \text{MSE}(\hat{c}(x, m), c(x, m))$$

where $\alpha$ is a weighting coefficient that balances performance and cost prediction.

For model selection approaches, we use cross-entropy loss:

$$\mathcal{L} = - \sum_{m \in \mathcal{M}} y_m \log(p(m|x))$$

where $y_m = 1$ if $m$ is the optimal model for query $x$ under the specified trade-off parameter $\lambda$, and 0 otherwise.

## D   SELECTION-BASED ROUTING EVALUATION

Selection based routing methods directly map queries to models without explicit utility scores, making construction of a full Pareto front challenging. We therefore evaluate these routers at three distinct cost-performance preferences and report the average utility scores across them. To enable consistent comparison across benchmarks with different cost scales, we normalize the trade-off parameter using $c_{\max}$, the maximum cost in each routing benchmark:

- **Low-cost preference** ($\lambda = 1.0/c_{\max}$): Heavily prioritizes efficiency while maintaining minimum performance requirements
- **Balanced preference** ($\lambda = 0.5/c_{\max}$): Balances performance and normalized cost
- **High-performance preference** ($\lambda = 0.1/c_{\max}$): Prioritizes response quality with reduced emphasis on efficiency

However, this evaluation approach has important limitations that can provide misleading assessments of routing quality. Since selection-based methods only report accuracy under fixed preference settings, they can obscure the true cost-performance trade-offs. High accuracy scores may simply reflect the selection of expensive models rather than intelligent routing decisions, as preference parameters only indirectly control cost through the utility function. This allows routers to achieve misleadingly high performance by consistently choosing costly but high-performing models.

To address these limitations, we recommend focusing primarily on utility prediction evaluation, which reveals the complete cost-performance relationship rather than single-point accuracy. For transparency and comparison with existing literature, we supplement selection-based results with actual cost-performance visualizations that reveal the true trade-offs achieved by different routing methods.

Table D.1 and D.2 present the utility scores averaged over 3 preference settings for text-based and multi-modal routing benchmarks, respectively.

For model selection approaches, there is no straightforward way to obtain the entire cost-quality Pareto front. Instead, we evaluate these routers using three distinct preference settings and report

Table D.1: Utility scores on a range of text routing benchmarks. All methods directly select the optimal routing model without explicitly estimating the utility scores. Scores are averaged over 3 preference settings. Higher is better.

| | AlpacaEval | | | HELM-Lite | | | OpenLLM | | | RouterBench | Avg |
|---|---|---|---|---|---|---|---|---|---|---|---|
| | OpenAI | Claude | Mistral | OpenAI | Claude | Google | LLaMA3 | Qwen2.5 | Yi1.5 | | |
| kNN (k=10) | 38.32 | 42.04 | 23.72 | 49.75 | 44.85 | 49.48 | 37.85 | 25.61 | 32.82 | 53.07 | 39.75 |
| kNN (k=100) | 37.89 | 43.99 | 23.88 | 48.93 | 41.86 | 48.63 | 37.56 | 22.61 | 32.09 | 52.36 | 38.98 |
| Linear | 54.61 | **51.14** | **30.96** | 48.53 | 42.13 | 48.69 | 38.43 | 24.38 | 32.11 | 52.36 | 42.33 |
| Linear (MF) | 54.54 | 51.14 | 30.96 | 48.59 | 42.57 | 48.73 | 38.57 | 24.82 | 32.15 | 52.36 | 42.44 |
| MLP | 50.81 | 51.14 | 30.52 | 49.30 | **45.37** | 48.95 | 39.11 | **26.20** | 33.30 | 53.80 | 42.85 |
| MLP (MF) | **54.71** | 51.14 | 30.96 | 48.60 | 43.55 | 48.72 | 38.58 | 24.62 | 32.24 | 52.36 | 42.55 |
| Graph (k=10) | 54.52 | 51.14 | 30.95 | 48.84 | 43.11 | 48.55 | 38.49 | 24.72 | 32.12 | 52.73 | 42.52 |
| Graph (k=100) | 54.61 | 51.14 | 30.96 | **50.47** | 44.62 | **50.72** | 37.41 | 21.69 | 32.09 | 52.28 | 42.60 |
| Attn (k=10) | 54.08 | 51.14 | 30.91 | 48.45 | 42.08 | 48.04 | 38.86 | 24.85 | 33.76 | 53.74 | 42.59 |
| Attn (k=100) | 54.04 | 51.14 | 30.96 | 49.13 | 44.31 | 49.17 | 38.51 | 23.42 | 33.93 | 54.43 | 42.90 |
| D-Attn (k=10) | 53.84 | 51.14 | 30.91 | 49.29 | 42.06 | 48.33 | **39.31** | 25.41 | **37.24** | **55.58** | **43.31** |
| D-Attn (k=100) | 53.55 | 51.14 | 30.96 | 48.95 | 40.76 | 48.60 | 37.95 | 23.94 | 34.44 | 52.90 | 42.32 |

Table D.2: Utility scores for vision language benchmark using selection based routing. Higher is better.

| | Blink | | Flickr30k | | MathVista | | MME | | MMMU | | Avg |
|---|---|---|---|---|---|---|---|---|---|---|---|
| | OpenAI | Claude | OpenAI | Claude | OpenAI | Claude | OpenAI | Claude | OpenAI | Claude | |
| kNN (k=10) | 60.42 | 71.13 | **56.88** | 47.50 | 34.48 | 24.69 | 77.41 | 67.74 | 52.08 | 49.53 | 54.19 |
| kNN (k=100) | 54.57 | 71.13 | 56.37 | 47.52 | 23.39 | 24.23 | 73.79 | 67.74 | 46.35 | 49.28 | 51.44 |
| Linear | 57.01 | 71.13 | 56.37 | 51.06 | 23.39 | 24.92 | 72.17 | 70.92 | 46.35 | 49.28 | 52.26 |
| Linear (MF) | 63.96 | 71.13 | 56.37 | 46.42 | 32.60 | 30.24 | 77.95 | 74.59 | 46.73 | 50.40 | 55.04 |
| MLP | **65.96** | **73.52** | 56.62 | 51.51 | **42.46** | 32.82 | 74.01 | 78.24 | **54.35** | **52.24** | **58.17** |
| MLP (MF) | 65.35 | 71.13 | 56.37 | 46.32 | 23.39 | 33.07 | 75.86 | 76.96 | 46.35 | 49.28 | 54.41 |
| Graph (k=10) | 54.80 | 71.07 | 56.37 | 48.69 | 32.11 | 24.70 | 73.09 | 72.60 | 46.22 | 49.12 | 52.88 |
| Graph (k=100) | 58.06 | 71.70 | 54.68 | 53.99 | 33.63 | 33.06 | 69.17 | 76.47 | 45.74 | 51.84 | 54.83 |
| Attn (k=10) | 62.01 | 71.30 | 56.88 | 56.42 | 39.10 | 28.21 | 77.56 | 82.39 | 54.22 | 51.53 | 57.96 |
| Attn (k=100) | 59.34 | 71.46 | 56.65 | 53.39 | 41.61 | 33.87 | 73.54 | 78.89 | 46.12 | 49.52 | 56.44 |
| D-Attn (k=10) | 60.16 | 70.32 | 55.29 | 56.19 | 30.42 | 29.75 | **78.53** | **83.75** | 47.71 | 51.27 | 56.34 |
| D-Attn (k=100) | 59.55 | 71.42 | 53.88 | 52.36 | 29.72 | **48.46** | 73.83 | 78.46 | 49.10 | 50.04 | 56.68 |

the final utility scores, following standard evaluation protocols in the literature. However, we must emphasize that reported utility scores alone can sometimes present an incomplete picture. The utility function combines both performance and cost in a weighted sum $(s(x,m) - \lambda \cdot c(x,m))$, but this single metric obscures the actual trade-off between these two dimensions. For instance, two routing approaches might achieve similar utility scores through different means—one by selecting higher-performing but costlier models, and another by choosing more cost-efficient models with slightly lower performance. Without visualizing the actual cost-performance points, it becomes difficult to understand these different strategies and their practical implications for deployment scenarios where budget constraints or performance requirements might vary.

To provide a more complete picture of this trade-off, we plot the actual cost-performance relationships in Fig.D.1 and Fig.D.2 for text-based and vision-language model routing benchmarks, respectively. These visualizations reveal the direct relationship between model cost and performance without the abstraction of a combined utility metric. The results demonstrate that simple kNN-based approaches remain highly competitive when compared to more complex routing methods operating under the same cost budget. In many cases, kNN routers achieve comparable or better performance than sophisticated neural architectures while maintaining similar cost efficiency, further supporting our core finding that simple, non-parametric routing methods offer a compelling alternative to complex learned approaches.

Notably, these plots often do not demonstrate a monotonic trend across the three preference settings, highlighting a significant limitation of the model selection formulation: the difficulty in precisely controlling the cost budget through preference parameter adjustment alone. This non-monotonic behavior underscores the challenge of balancing performance and cost when routing decisions are made directly, rather than through explicit utility estimation, and further motivates our parallel investigation of utility prediction approaches that allow for more flexible exploration of the entire Pareto front.

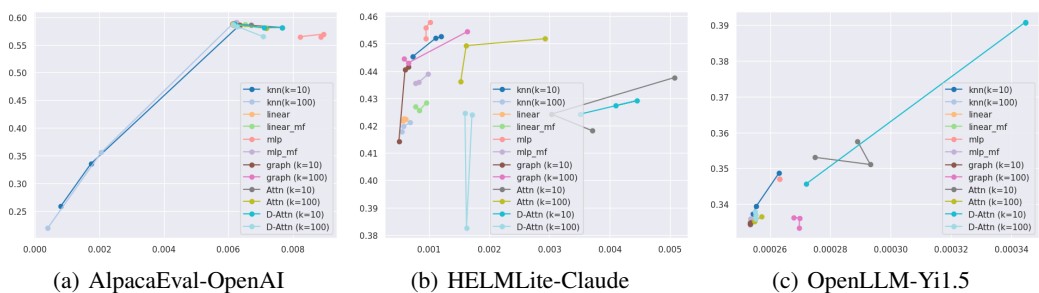

(a) AlpacaEval-OpenAI      (b) HELMLite-Claude      (c) OpenLLM-Yi1.5

Figure D.1: Cost-Quality tradeoff for text-based routing benchmarks using model selection approaches.

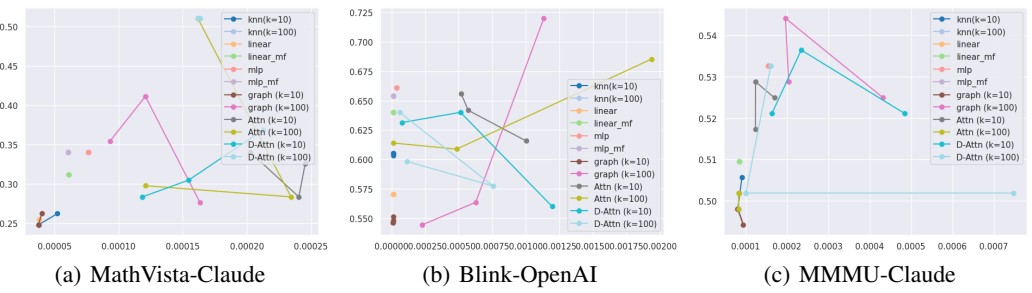

(a) MathVista-Claude      (b) Blink-OpenAI      (c) MMMU-Claude

Figure D.2: Cost-Quality tradeoff for VLM routing benchmarks using model selection approaches.

# E  DETAILED RESULTS

In the main text, we report RouterBench results as averages across all six datasets. For a more granular analysis, Tables E.1 and E.2 provide the detailed performance breakdown for each individual dataset under the utility prediction and model selection formulations, respectively. Additionally, Tables E.3 and E.4 present detailed results across three different preference settings for the text-based and multi-modal (VLM) routing benchmarks.

Table E.1: AUC score on RouterBench using utility prediction routing. Higher is better.

|  | Arcc | GSM | MBPP | MMLU | Hellaswag | Winogrande | Avg |
|---|---|---|---|---|---|---|---|
| Oracle | 97.99 | 74.59 | 85.02 | 96.44 | 97.92 | 99.48 | 91.91 |
| Random | 64.67 | 52.47 | 50.74 | 57.41 | 54.74 | 49.55 | 54.93 |
| kNN (k=10) | 88.15 | 63.82 | 60.16 | 73.91 | 87.93 | 71.35 | 74.22 |
| kNN (k=100) | 91.80 | 64.72 | 58.67 | 80.81 | 89.39 | 77.91 | 77.22 |
| Linear | 92.27 | 65.50 | 60.76 | 81.05 | 87.87 | 78.63 | 77.68 |
| Linear (MF) | 91.80 | 64.55 | 60.08 | 80.91 | 88.07 | 78.18 | 77.27 |
| MLP | 91.78 | 65.03 | 59.35 | 80.22 | 87.85 | 78.22 | 77.08 |
| MLP (MF) | 91.70 | 65.38 | 60.05 | 80.94 | 87.85 | 78.50 | 77.40 |
| Graph (k=10) | 91.70 | 62.23 | 56.36 | 80.76 | 87.89 | 78.52 | 76.24 |
| Graph (k=100) | 91.39 | 62.76 | 59.38 | 80.88 | 87.89 | 78.93 | 76.87 |
| Attn (k=10) | 89.49 | 62.08 | 56.36 | 72.12 | 87.88 | 73.26 | 73.53 |
| Attn (k=100) | 91.63 | 64.07 | 60.85 | 81.01 | 87.86 | 78.82 | 77.37 |
| D-Attn (k=10) | 89.43 | 62.51 | 58.67 | 74.15 | 87.91 | 73.72 | 74.40 |
| D-Attn (k=100) | 91.67 | 64.48 | 60.14 | 80.71 | 89.40 | 78.46 | 77.48 |

Table E.2: Utility score on RouterBench using selection based routing. Higher is better.

|  |  | Arcc | GSM | MBPP | MMLU | Hellaswag | Winogrande | Avg |
|---|---|---|---|---|---|---|---|---|
| High-Performance | kNN (k=10) | 64.75 | 50.30 | 36.41 | 55.31 | 59.74 | 51.95 | 53.08 |
|  | kNN (k=100) | 64.53 | 50.89 | 33.32 | 53.39 | 59.74 | 52.49 | 52.39 |
|  | Linear | 64.53 | 50.86 | 33.32 | 53.39 | 59.74 | 52.49 | 52.39 |
|  | Linear (MF) | 64.53 | 50.88 | 33.32 | 53.39 | 59.74 | 52.49 | 52.39 |
|  | MLP | 64.53 | 51.40 | 42.60 | 53.39 | 59.74 | 52.47 | 54.02 |
|  | MLP (MF) | 64.53 | 50.86 | 33.32 | 53.39 | 59.74 | 52.49 | 52.39 |
|  | Graph (k=10) | 64.53 | 50.90 | 32.54 | 53.39 | 59.74 | 53.27 | 52.40 |
|  | Graph (k=100) | 65.95 | 50.90 | 31.73 | 54.76 | 59.74 | 57.21 | 53.38 |
|  | Attn (k=10) | 67.32 | 50.64 | 37.85 | 56.49 | 59.74 | 54.03 | 54.35 |
|  | Attn (k=100) | 64.35 | 51.18 | 41.57 | 55.80 | 59.74 | 36.20 | 51.47 |
|  | D-Attn (k=10) | 66.35 | 44.22 | 35.53 | 46.06 | 59.74 | 50.64 | 50.42 |
|  | D-Attn (k=100) | 65.13 | 51.05 | 32.51 | 54.77 | 25.57 | 54.02 | 47.18 |
| Balanced | kNN (k=10) | 64.53 | 50.23 | 36.31 | 55.06 | 59.74 | 52.49 | 53.06 |
|  | kNN (k=100) | 64.53 | 50.78 | 33.25 | 53.39 | 59.74 | 52.49 | 52.36 |
|  | Linear | 64.53 | 50.73 | 33.25 | 53.39 | 59.74 | 52.49 | 52.36 |
|  | Linear (MF) | 64.53 | 50.74 | 33.25 | 53.39 | 59.74 | 52.49 | 52.36 |
|  | MLP | 64.53 | 51.01 | 42.46 | 53.39 | 59.74 | 52.38 | 53.92 |
|  | MLP (MF) | 64.53 | 50.76 | 33.25 | 53.39 | 59.74 | 52.49 | 52.36 |
|  | Graph (k=10) | 64.53 | 50.78 | 34.02 | 53.39 | 59.74 | 58.34 | 53.47 |
|  | Graph (k=100) | 65.38 | 41.45 | 29.23 | 53.39 | 59.74 | 53.58 | 50.46 |
|  | Attn (k=10) | 67.40 | 50.12 | 33.94 | 56.00 | 59.74 | 51.91 | 53.19 |
|  | Attn (k=100) | 60.53 | 50.73 | 41.61 | 55.58 | 84.10 | 51.49 | 57.34 |
|  | D-Attn (k=10) | 84.52 | 49.94 | 58.29 | 52.09 | 84.10 | 52.06 | 63.50 |
|  | D-Attn (k=100) | 73.40 | 50.54 | 33.11 | 54.44 | 54.37 | 54.42 | 53.38 |
| Low-Cost | kNN (k=10) | 64.52 | 50.45 | 36.19 | 54.98 | 59.74 | 52.48 | 53.06 |
|  | kNN (k=100) | 64.52 | 50.63 | 33.17 | 53.39 | 59.74 | 52.48 | 52.32 |
|  | Linear | 64.52 | 50.59 | 33.17 | 53.39 | 59.74 | 52.48 | 52.32 |
|  | Linear (MF) | 64.52 | 50.59 | 33.17 | 53.39 | 59.74 | 52.48 | 52.32 |
|  | MLP | 64.52 | 50.85 | 39.96 | 53.39 | 59.74 | 52.27 | 53.46 |
|  | MLP (MF) | 64.52 | 50.60 | 33.17 | 53.39 | 59.74 | 52.48 | 52.32 |
|  | Graph (k=10) | 64.52 | 50.63 | 33.17 | 53.39 | 59.74 | 52.47 | 52.32 |
|  | Graph (k=100) | 64.80 | 50.63 | 32.94 | 53.39 | 59.74 | 56.43 | 52.99 |
|  | Attn (k=10) | 64.52 | 49.12 | 36.63 | 56.35 | 59.74 | 55.65 | 53.67 |
|  | Attn (k=100) | 67.30 | 50.56 | 37.67 | 53.33 | 59.74 | 58.24 | 54.47 |
|  | D-Attn (k=10) | 64.76 | 49.57 | 37.19 | 53.50 | 59.74 | 52.21 | 52.83 |
|  | D-Attn (k=100) | 63.64 | 59.87 | 56.93 | 53.91 | 59.74 | 54.80 | 58.15 |

# F    LATENCY ANALYSIS

In this section, we analyze the latency of each routing approach on Routerbench. Table F.1 reports the inference time required to route all examples in each test set. This analysis provides insights into the computational efficiency of different routing approaches, which is an important consideration for practical deployment.

We measured the total processing time for each routing method across all datasets, excluding the one-time costs of building nearest neighbor indices and training predictive models. For a fair comparison, CPU-based methods (kNN, Linear, and MLP predictors) were run on an Intel(R) Xeon(R) Platinum 8275CL processor, while GPU-accelerated methods (Graph and attention-based predictors) were executed on an NVIDIA A100 GPU. The kNN predictor leverages ScaNN for efficient nearest neighbor search, enabling fast retrieval even with larger support sets. Linear and MLP predictors are implemented using the sklearn package.

The results demonstrate that simpler methods offer substantial computational advantages. kNN-based approaches are remarkably efficient, with kNN (k=100) requiring only 10.95 seconds on average per dataset, making it the fastest method overall. Simple parametric models (Linear and MLP variants) show moderate processing times (14-19 seconds). In contrast, graph and attention-based

Table E.3: Utility scores on a range of text routing benchmarks. All methods directly select the optimal routing model without explicitly estimating the utility scores. Higher is better.

| | | AlpacaEval | | | HELM-Lite | | | OpenLLM | | |
|---|---|---|---|---|---|---|---|---|---|---|
| | | OpenAI | Claude | Mistral | OpenAI | Claude | Google | LLaMA3 | Qwen2.5 | Yi1.5 |
| High-Performance | kNN (k=10) | 57.72 | 53.38 | 27.61 | 49.83 | 45.23 | 49.56 | 39.81 | 32.12 | 34.61 |
| | kNN (k=100) | 58.03 | 53.38 | 26.81 | 49.00 | 42.09 | 48.69 | 39.30 | 25.36 | 33.19 |
| | Linear | 58.01 | 53.38 | 33.24 | 48.57 | 42.23 | 48.76 | 39.03 | 24.73 | 33.20 |
| | Linear (MF) | 57.85 | 53.38 | 33.29 | 48.66 | 42.81 | 48.87 | 39.17 | 25.19 | 33.25 |
| | MLP | 55.33 | 53.38 | 33.22 | 49.38 | 45.76 | 48.93 | 39.78 | 26.65 | 34.44 |
| | MLP (MF) | 58.29 | 53.38 | 33.26 | 48.69 | 43.86 | 48.83 | 39.19 | 24.99 | 33.34 |
| | Graph (k=10) | 57.82 | 53.38 | 33.21 | 48.86 | 41.42 | 48.59 | 39.34 | 24.35 | 33.19 |
| | Graph (k=100) | 58.03 | 53.38 | 33.23 | 48.65 | 45.39 | 51.10 | 37.41 | 21.57 | 33.36 |
| | Attn (k=10) | 57.16 | 53.38 | 33.27 | 49.06 | 43.61 | 48.85 | 40.20 | 24.95 | 35.46 |
| | Attn (k=100) | 58.03 | 53.38 | 33.23 | 49.47 | 45.09 | 49.94 | 38.97 | 21.57 | 33.40 |
| | D-Attn (k=10) | 57.16 | 53.38 | 33.27 | 47.84 | 42.61 | 48.45 | 39.95 | 26.89 | 38.72 |
| | D-Attn (k=100) | 55.57 | 53.38 | 33.23 | 47.90 | 42.42 | 49.34 | 39.33 | 22.43 | 33.45 |
| Balanced | kNN (k=10) | 32.41 | 44.93 | 23.23 | 49.76 | 45.03 | 49.45 | 37.31 | 23.15 | 32.66 |
| | kNN (k=100) | 34.23 | 51.11 | 25.34 | 48.92 | 41.88 | 48.64 | 37.09 | 21.63 | 32.16 |
| | Linear | 54.88 | 51.31 | 30.98 | 48.53 | 42.08 | 48.68 | 38.48 | 24.40 | 32.19 |
| | Linear (MF) | 54.85 | 51.31 | 30.97 | 48.58 | 42.42 | 48.72 | 38.61 | 24.85 | 32.24 |
| | MLP | 51.19 | 51.31 | 30.29 | 49.24 | 45.45 | 49.04 | 39.16 | 26.23 | 33.39 |
| | MLP (MF) | 54.91 | 51.31 | 30.98 | 48.58 | 43.46 | 48.70 | 38.63 | 24.65 | 32.33 |
| | Graph (k=10) | 54.88 | 51.31 | 30.98 | 48.87 | 43.95 | 48.55 | 38.50 | 25.69 | 32.22 |
| | Graph (k=100) | 54.88 | 51.31 | 30.98 | 50.14 | 44.20 | 51.21 | 38.80 | 21.82 | 32.26 |
| | Attn (k=10) | 54.24 | 51.31 | 30.98 | 48.05 | 41.96 | 47.75 | 38.47 | 25.11 | 33.64 |
| | Attn (k=100) | 53.39 | 51.31 | 30.98 | 48.88 | 44.68 | 48.77 | 38.66 | 24.62 | 32.25 |
| | D-Attn (k=10) | 53.51 | 51.31 | 30.98 | 48.04 | 42.23 | 48.42 | 39.64 | 25.09 | 37.36 |
| | D-Attn (k=100) | 54.31 | 51.31 | 30.98 | 49.71 | 38.01 | 47.86 | 38.46 | 24.97 | 32.32 |
| Low-Cost | kNN (k=10) | 24.83 | 27.82 | 20.31 | 49.66 | 44.30 | 49.42 | 36.44 | 21.56 | 31.20 |
| | kNN (k=100) | 21.41 | 27.47 | 19.49 | 48.88 | 41.60 | 48.57 | 36.29 | 20.84 | 30.91 |
| | Linear | 50.93 | 48.72 | 28.66 | 48.49 | 42.08 | 48.62 | 37.78 | 24.00 | 30.93 |
| | Linear (MF) | 50.93 | 48.72 | 28.62 | 48.53 | 42.47 | 48.61 | 37.92 | 24.43 | 30.97 |
| | MLP | 45.91 | 48.72 | 28.06 | 49.28 | 44.90 | 48.87 | 38.40 | 25.71 | 32.08 |
| | MLP (MF) | 50.93 | 48.72 | 28.63 | 48.52 | 43.32 | 48.63 | 37.93 | 24.23 | 31.06 |
| | Graph (k=10) | 50.85 | 48.72 | 28.66 | 48.79 | 43.95 | 48.50 | 37.62 | 24.11 | 30.94 |
| | Graph (k=100) | 50.93 | 48.72 | 28.66 | 52.62 | 44.28 | 49.85 | 36.02 | 21.69 | 30.64 |
| | Attn (k=10) | 50.85 | 48.72 | 28.49 | 48.24 | 40.68 | 47.51 | 37.90 | 24.50 | 32.18 |
| | Attn (k=100) | 50.69 | 48.72 | 28.66 | 49.03 | 43.16 | 48.79 | 37.89 | 24.08 | 36.13 |
| | D-Attn (k=10) | 50.85 | 48.72 | 28.49 | 52.00 | 41.35 | 48.13 | 38.35 | 24.25 | 35.63 |
| | D-Attn (k=100) | 50.77 | 48.72 | 28.66 | 49.24 | 41.86 | 48.60 | 36.07 | 24.41 | 37.56 |

methods are approximately 13-14× slower than kNN, with average processing times exceeding 144 seconds per dataset.

These findings further strengthen our argument for simpler routing approaches, highlighting that kNN not only matches or exceeds the routing performance of more complex methods but also offers significant advantages in computational efficiency. For large-scale deployment scenarios where thousands of routing decisions may be needed per second, these latency differences become particularly important.

# G  OUT-OF-DISTRIBUTION QUERIES

To evaluate the generalization ability of routing models under distribution shift, we conduct a comprehensive cross-dataset evaluation. For each of the six datasets in RouterBench, we train models on one dataset and evaluate on all others, measuring AUC scores as our primary metric. This results in 36 (train, test) pairs per model.

We visualize the results in a test-centric manner: for each test dataset, we compare in-distribution (ID) performance—when the train and test datasets are identical—with out-of-distribution (OOD) performance, where models are trained on different datasets. Figure G.1 presents the AUC scores for each test dataset across different training datasets, while Table G.1 reports the ID and OOD performance averaged across all six test datasets.

As expected, models consistently achieve higher AUC scores when evaluated in-distribution. However, the degree of generalization to OOD test sets varies considerably by model architecture and training dataset. For instance, models trained on hellaswag generalize well to mmlu, but per-

Table E.4: Utility scores for vision language benchmark using selection based routing. Higher is better.

| | | Blink | | Flickr30k | | MathVista | | MME | | MMMU | |
|---|---|---|---|---|---|---|---|---|---|---|---|
| | | OpenAI | Claude | OpenAI | Claude | OpenAI | Claude | OpenAI | Claude | OpenAI | Claude |
| High-Performance | kNN (k=10) | 60.51 | 72.26 | 57.11 | 49.31 | 37.58 | 26.09 | 78.23 | 69.50 | 52.10 | 50.46 |
| | kNN (k=100) | 54.60 | 72.26 | 56.45 | 49.57 | 23.40 | 24.71 | 74.76 | 69.50 | 46.36 | 49.71 |
| | Linear | 57.04 | 72.26 | 56.45 | 53.03 | 23.40 | 25.42 | 72.21 | 72.92 | 46.36 | 49.71 |
| | Linear (MF) | 63.99 | 72.26 | 56.45 | 46.64 | 32.62 | 31.02 | 78.00 | 76.99 | 46.74 | 50.85 |
| | MLP | 66.06 | 75.68 | 57.73 | 54.64 | 42.54 | 33.81 | 74.06 | 81.25 | 54.40 | 53.07 |
| | MLP (MF) | 65.38 | 72.26 | 56.45 | 47.54 | 23.40 | 33.86 | 75.91 | 79.69 | 46.36 | 49.71 |
| | Graph (k=10) | 54.60 | 72.26 | 56.45 | 46.50 | 34.03 | 26.12 | 73.83 | 73.37 | 46.36 | 49.71 |
| | Graph (k=100) | 54.30 | 73.58 | 48.02 | 61.94 | 25.47 | 27.17 | 69.67 | 79.13 | 45.15 | 51.96 |
| | Attn (k=10) | 60.98 | 72.60 | 56.51 | 62.30 | 35.37 | 31.89 | 77.25 | 88.02 | 53.96 | 52.28 |
| | Attn (k=100) | 61.38 | 71.38 | 56.06 | 54.94 | 43.92 | 50.58 | 67.66 | 83.94 | 47.12 | 50.09 |
| | D-Attn (k=10) | 55.30 | 71.75 | 51.62 | 61.90 | 41.01 | 36.24 | 55.73 | 88.02 | 48.48 | 51.51 |
| | D-Attn (k=100) | 63.96 | 72.56 | 48.02 | 52.66 | 29.77 | 50.58 | 72.36 | 85.73 | 45.58 | 49.27 |
| Balanced | kNN (k=10) | 60.48 | 71.22 | 56.97 | 47.15 | 35.42 | 24.26 | 77.03 | 67.87 | 52.08 | 49.31 |
| | kNN (k=100) | 54.58 | 71.22 | 56.37 | 47.29 | 23.39 | 24.26 | 73.33 | 67.87 | 46.35 | 49.31 |
| | Linear | 57.01 | 71.22 | 56.37 | 51.53 | 23.39 | 24.96 | 72.17 | 71.07 | 46.35 | 49.31 |
| | Linear (MF) | 63.96 | 71.22 | 56.37 | 45.48 | 32.60 | 30.30 | 77.95 | 74.78 | 46.73 | 50.44 |
| | MLP | 65.97 | 73.68 | 56.77 | 51.59 | 42.47 | 32.90 | 74.02 | 78.47 | 54.35 | 52.31 |
| | MLP (MF) | 65.35 | 71.22 | 56.37 | 46.06 | 23.39 | 33.13 | 75.87 | 77.17 | 46.35 | 49.31 |
| | Graph (k=10) | 55.10 | 71.03 | 56.37 | 53.52 | 29.05 | 24.26 | 71.71 | 75.01 | 45.96 | 48.84 |
| | Graph (k=100) | 54.52 | 71.61 | 58.28 | 49.86 | 33.77 | 39.33 | 70.76 | 78.30 | 45.21 | 53.20 |
| | Attn (k=10) | 62.51 | 70.98 | 57.25 | 56.62 | 44.50 | 24.77 | 78.27 | 79.73 | 48.40 | 52.11 |
| | Attn (k=100) | 59.45 | 72.36 | 57.09 | 56.76 | 40.86 | 24.86 | 75.87 | 78.80 | 46.00 | 49.30 |
| | D-Attn (k=10) | 62.50 | 71.85 | 56.71 | 56.55 | 26.45 | 28.19 | 90.22 | 83.60 | 48.79 | 52.20 |
| | D-Attn (k=100) | 55.53 | 71.28 | 56.90 | 56.63 | 36.05 | 48.62 | 69.26 | 78.19 | 52.68 | 49.57 |
| Low-Cost | kNN (k=10) | 60.27 | 69.92 | 56.55 | 46.03 | 30.43 | 23.71 | 76.97 | 65.84 | 52.05 | 48.81 |
| | kNN (k=100) | 54.54 | 69.92 | 56.28 | 45.69 | 23.38 | 23.71 | 73.27 | 65.84 | 46.34 | 48.81 |
| | Linear | 56.98 | 69.92 | 56.28 | 48.62 | 23.38 | 24.39 | 72.12 | 68.76 | 46.33 | 48.81 |
| | Linear (MF) | 63.93 | 69.92 | 56.28 | 47.14 | 32.57 | 29.39 | 77.90 | 72.01 | 46.71 | 49.92 |
| | MLP | 65.86 | 71.19 | 55.35 | 48.31 | 42.38 | 31.76 | 73.96 | 74.99 | 54.30 | 51.35 |
| | MLP (MF) | 65.31 | 69.92 | 56.28 | 45.35 | 23.38 | 32.23 | 75.81 | 74.02 | 46.34 | 48.81 |
| | Graph (k=10) | 54.71 | 69.92 | 56.28 | 46.05 | 33.25 | 23.71 | 73.73 | 69.41 | 46.34 | 48.81 |
| | Graph (k=100) | 65.37 | 69.92 | 57.73 | 50.17 | 41.66 | 32.67 | 67.08 | 71.99 | 46.86 | 50.36 |
| | Attn (k=10) | 62.53 | 70.31 | 56.88 | 50.33 | 37.42 | 27.98 | 77.17 | 79.41 | 60.31 | 50.20 |
| | Attn (k=100) | 57.20 | 70.65 | 56.81 | 48.47 | 40.05 | 26.17 | 77.10 | 73.93 | 45.24 | 49.16 |
| | D-Attn (k=10) | 62.69 | 67.35 | 57.54 | 50.12 | 23.80 | 24.83 | 89.63 | 79.62 | 45.85 | 50.09 |
| | D-Attn (k=100) | 59.16 | 70.43 | 56.71 | 47.78 | 23.35 | 46.17 | 79.86 | 71.47 | 49.04 | 51.28 |

Table F.1: The cumulative time to process the each test set.

| | arcc | gsm | hellaswag | mbpp | mmlu | winogrande | AVG | SUM |
|---|---|---|---|---|---|---|---|---|
| **kNN (k=10)** | 2.48 | 12.81 | 18.64 | 0.74 | 37.51 | 3.58 | 12.62 | 75.76 |
| **kNN (k=100)** | 2.69 | 13.75 | 19.37 | 0.80 | 26.73 | 2.35 | 10.95 | 65.69 |
| **Linear** | 3.56 | 18.30 | 24.27 | 1.07 | 34.15 | 3.16 | 14.09 | 84.51 |
| **Linear (MF)** | 4.63 | 22.99 | 20.07 | 1.60 | 43.69 | 2.73 | 15.95 | 95.71 |
| **MLP** | 4.25 | 35.76 | 28.34 | 1.23 | 43.29 | 3.57 | 19.41 | 116.44 |
| **MLP (MF)** | 4.86 | 15.63 | 20.99 | 1.20 | 45.51 | 4.33 | 15.42 | 92.52 |
| **Graph (k=10)** | 32.37 | 186.73 | 250.98 | 8.78 | 359.70 | 27.78 | 144.39 | 866.34 |
| **Graph (k=100)** | 33.51 | 186.48 | 253.32 | 8.82 | 361.71 | 28.19 | 145.34 | 872.03 |
| **Attn (k=10)** | 32.80 | 186.96 | 256.73 | 9.08 | 363.43 | 28.64 | 146.27 | 877.64 |
| **Attn (k=100)** | 33.68 | 189.76 | 256.94 | 8.94 | 365.37 | 28.62 | 147.22 | 883.31 |
| **D-Attn (k=10)** | 33.92 | 193.12 | 264.91 | 9.30 | 375.58 | 28.68 | 150.92 | 905.51 |
| **D-Attn (k=100)** | 33.91 | 194.28 | 264.21 | 9.04 | 375.98 | 28.90 | 151.05 | 906.32 |

form poorly on `mbpp`. Conversely, models trained on `gsm` demonstrate strong performance on `mbpp`, while struggling with `mmlu`. We attribute these patterns to varying degrees of domain shift between the datasets.

On average, all routing approaches show performance degradation when evaluating on OOD queries, though kNN-based approaches exhibit more moderate decreases compared to their more complex counterparts. The simple kNN (k=100) model shows the smallest performance drop (2.63 points), while Linear (MF) experiences the largest degradation (6.67 points). These findings highlight that simpler models may be more robust to distribution shifts in routing tasks.

Overall, no single model universally dominates across all cross-dataset evaluations, emphasizing the importance of data diversity and robustness-aware model design for effective routing decisions.

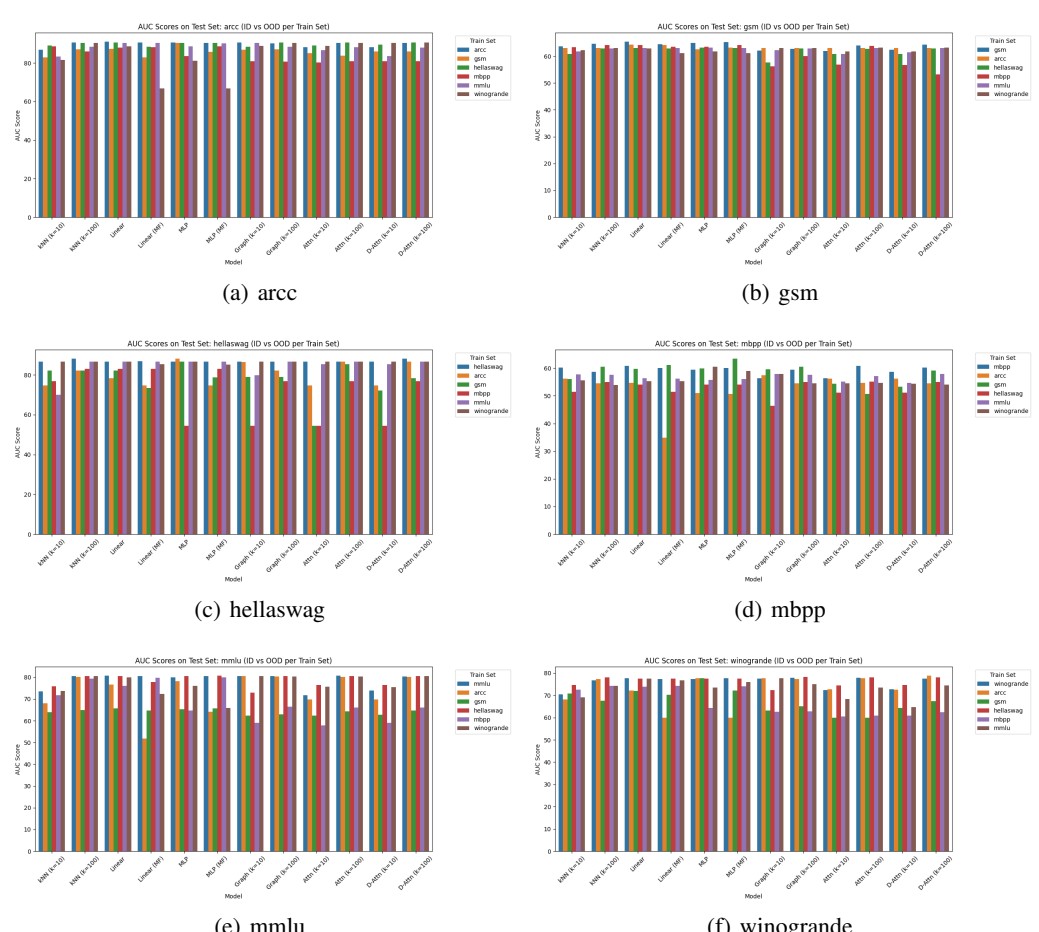

Figure G.1: AUC scores for each testset when routers are trained on different trainset.

Table G.1: Average AUC scores across all test sets.

| Model | Avg_ID | Avg_OOD | Delta |
|---|---|---|---|
| kNN (k=10) | 73.53 | 70.41 | 3.13 |
| kNN (k=100) | 76.54 | 73.91 | 2.63 |
| Linear | 77.03 | 73.70 | 3.33 |
| Linear (MF) | 76.63 | 69.96 | 6.67 |
| MLP | 76.45 | 72.23 | 4.22 |
| MLP (MF) | 76.77 | 71.47 | 5.30 |
| Graph (k=10) | 75.58 | 70.39 | 5.19 |
| Graph (k=100) | 76.20 | 72.38 | 3.82 |
| Attn (k=10) | 72.87 | 67.93 | 4.94 |
| Attn (k=100) | 76.71 | 72.19 | 4.52 |
| D-Attn (k=10) | 73.74 | 68.58 | 5.16 |
| D-Attn (k=100) | 76.81 | 72.32 | 4.49 |

# H EMBEDDING ANALYSIS

Table H.1 examines the impact of different embedding models on routing performance. Switching from BERT to SFR embeddings provides modest improvements across most methods, with linear and MLP models showing the largest gains. Importantly, the relative ranking of routing methods

remains similar across embedding types—simple methods maintain their competitive advantage regardless of the underlying representation quality.

The consistent importance of neighborhood size (k) across different embeddings reinforces that the locality properties we observe are fundamental to the routing problem rather than artifacts of particular representation spaces.

Table H.1: Ablation study on prompt embedding models.

| | | Arcc | GSM | MBPP | MMLU | Hellaswag | Winogrande | Avg |
|---|---|---|---|---|---|---|---|---|
| BERT Embedding | kNN (k=10) | 88.15 | 63.82 | 60.16 | 73.91 | 87.93 | 71.35 | 74.22 |
| | kNN (k=100) | 91.80 | 64.72 | 58.67 | 80.81 | 89.39 | 77.91 | 77.22 |
| | Linear | 92.27 | 65.50 | 60.76 | 81.05 | 87.87 | 78.63 | **77.68** |
| | Linear (MF) | 91.80 | 64.55 | 60.08 | 80.91 | 88.07 | 78.18 | 77.27 |
| | MLP | 91.78 | 65.03 | 59.35 | 80.22 | 87.85 | 78.22 | 77.08 |
| | MLP (MF) | 91.70 | 65.38 | 60.05 | 80.94 | 87.85 | 78.50 | 77.40 |
| | Graph (k=10) | 91.70 | 62.23 | 56.36 | 80.76 | 87.89 | 78.52 | 76.24 |
| | Graph (k=100) | 91.39 | 62.76 | 59.38 | 80.88 | 87.89 | 78.93 | 76.87 |
| | Attn (k=10) | 89.49 | 62.08 | 56.36 | 72.12 | 87.88 | 73.26 | 73.53 |
| | Attn (k=100) | 91.63 | 64.07 | 60.85 | 81.01 | 87.86 | 78.82 | 77.37 |
| | D-Attn (k=10) | 89.43 | 62.51 | 58.67 | 74.15 | 87.91 | 73.72 | 74.40 |
| | D-Attn (k=100) | 91.67 | 64.48 | 60.14 | 80.71 | 89.40 | 78.46 | 77.48 |
| SFR Embedding | kNN (k=10) | 88.08 | 63.88 | 61.18 | 74.61 | 84.78 | 73.08 | 74.27 |
| | kNN (k=100) | 91.33 | 64.95 | 62.22 | 80.94 | 89.08 | 77.76 | 77.71 |
| | Linear | 92.02 | 65.60 | 64.68 | 80.30 | 89.41 | 78.42 | **78.41** |
| | Linear (MF) | 90.80 | 64.71 | 61.72 | 80.01 | 89.43 | 78.62 | 77.55 |
| | MLP | 90.05 | 62.92 | 63.91 | 80.43 | 87.75 | 77.04 | 77.02 |
| | MLP (MF) | 91.86 | 65.16 | 64.64 | 80.67 | 89.42 | 78.43 | 78.36 |
| | Graph (k=10) | 91.04 | 59.62 | 56.37 | 80.92 | 87.97 | 78.77 | 75.78 |
| | Graph (k=100) | 91.45 | 61.52 | 60.95 | 81.01 | 88.95 | 78.56 | 77.07 |
| | Attn (k=10) | 88.13 | 62.13 | 57.10 | 76.56 | 86.56 | 74.18 | 74.11 |
| | Attn (k=100) | 91.32 | 64.24 | 60.11 | 80.97 | 89.04 | 78.47 | 77.36 |
| | D-Attn (k=10) | 88.16 | 62.41 | 56.24 | 76.48 | 86.68 | 73.38 | 73.89 |
| | D-Attn (k=100) | 91.11 | 64.34 | 59.33 | 81.02 | 89.18 | 78.57 | 77.26 |

# I   PROOF

**Theorem 1.** *For a query distribution $\mathcal{D}$ with $\delta$-locality in utility space:*

*(a) A kNN router requires a training sample size of $\Theta\left(\frac{C_{\mathcal{X},d}}{\delta^d} \log\left(\frac{1}{\alpha}\right)\right)$ to achieve expected regret $O(\epsilon(\delta))$ with probability $1 - \alpha$, where $d$ is the intrinsic dimension of the embedding space and $C_{\mathcal{X},d}$ is a constant depending on the space.*

*(b) A parametric router with $L$ Lipschitz-continuous layers requires a training sample size of $\Omega(L/\epsilon(\delta)^2)$ to achieve the same regret bound.*

*Proof.* We first derive the sample complexity for kNN and parametric approaches and then compare them.

**Part 1: kNN Router Sample Complexity**

Let $\mathcal{X}$ be the query embedding space with intrinsic dimension $d$ and $u : \mathcal{X} \times \mathcal{M} \to \mathbb{R}$ be the true utility function mapping query-model pairs to utility scores. By the $\delta$-locality property, for any two queries $x_1, x_2$ with $d(x_1, x_2) < \delta$, we have $|u(x_1, m) - u(x_2, m)| < \epsilon(\delta)$ for all models $m \in \mathcal{M}$.

**Step 1:** First, we establish what makes a kNN estimate accurate.

For a query $x$, let $\hat{u}(x, m)$ be the kNN estimate of model $m$'s utility:

$$\hat{u}(x, m) = \frac{1}{k} \sum_{x_i \in \mathcal{N}_k(x)} u(x_i, m),$$

where $\mathcal{N}_k(x)$ is the set of $k$ nearest neighbors of $x$ in the training set. By the $\delta$-locality property, if all $k$ neighbors are within distance $\delta$ of $x$, then the approximation error is bounded by $\epsilon(\delta)$:

$$|\hat{u}(x, m) - u(x, m)| < \epsilon(\delta).$$

**Step 2:** We analyze the sample complexity needed to ensure sufficient neighborhood coverage.

Let $N(\mathcal{X}, \delta/2)$ be the $\delta/2$-covering number of $\mathcal{X}$, which is the minimum number of balls of radius $\delta/2$ needed to cover $\mathcal{X}$. For a space with intrinsic dimension $d$, this covering number scales as $N(\mathcal{X}, \delta/2) = \Theta(C_{\mathcal{X},d}/\delta^d)$, where $C_{\mathcal{X},d}$ is a constant depending on the properties of $\mathcal{X}$ (Kolmogorov & Tikhomirov, 1959; Luxburg & Bousquet, 2004).

Let's divide the space into $N(\mathcal{X}, \delta/2)$ regions corresponding to this cover. If we ensure that each region contains at least $k$ training points, then any query will have at least $k$ neighbors within distance $\delta$.

**Step 3:** We compute the sample size needed to populate each region with enough points.

Assume we draw $n$ training samples i.i.d. from distribution $\mathcal{D}$. For a region $R_i$ with probability mass $\mathcal{D}(R_i)$, the probability that fewer than $k$ samples fall into $R_i$ is:

$$P(\text{fewer than } k \text{ samples in } R_i) = \sum_{j=0}^{k-1} \binom{n}{j} \mathcal{D}(R_i)^j (1 - \mathcal{D}(R_i))^{n-j}.$$

We need to ensure that with high probability, $X_i \geq k$ for all regions.

For a single region $R_i$, using the Chernoff bound for the lower tail of a binomial distribution with mean $\mu = nD(R_i)$:

$$P(X_i \leq (1-t)\mu) \leq \exp(-t^2\mu/2) \quad \text{for any } 0 < t < 1$$

Setting $(1-t)nD(R_i) = k$, which implies $t = 1 - \frac{k}{nD(R_i)}$, and imposing the constraint $nD(R_i) \geq 2k$ (ensuring $t \geq \frac{1}{2}$), we get:

$$P(X_i < k) \leq \exp(-t^2 nD(R_i)/2) \leq \exp\left(-\frac{(1/4)nD(R_i)}{2}\right) = \exp\left(-\frac{nD(R_i)}{8}\right)$$

For the union bound to work across all $N(X, \delta/2)$ regions, we need:

$$P(X_i < k) \leq \frac{\alpha}{N(X, \delta/2)} \text{ for each region } R_i$$

Therefore:

$$\exp\left(-\frac{nD(R_i)}{8}\right) \leq \frac{\alpha}{N(X, \delta/2)}$$

Taking logarithms:

$$-\frac{nD(R_i)}{8} \leq \ln(\alpha) - \ln(N(X, \delta/2))$$

$$nD(R_i) \geq 8\ln\left(\frac{1}{\alpha}\right) + 8\ln\left(N(X, \delta/2)\right)$$

To maintain consistency with our constraint $nD(R_i) \geq 2k$, we require:

$$2k \geq 8\ln\left(\frac{1}{\alpha}\right) + 8\ln\left(N(X, \delta/2)\right)$$

$$k \geq 4\ln\left(\frac{1}{\alpha}\right) + 4\ln\left(N(X, \delta/2)\right)$$

To satisfy $nD(R_i) \geq 2k$ for all regions, we need:

$$n \geq \frac{2k}{\min_i D(R_i)}$$

Under a reasonable assumption that the distribution $\mathcal{D}$ doesn't assign extremely small probability to any significant region (specifically, $\min_i D(R_i) \geq \frac{c}{N(X,\delta/2)}$ for some constant $c > 0$), we get:

$$n \geq \frac{2kN(X,\delta/2)}{c} = \Theta\left(\frac{k \cdot C_{X,d}}{c \cdot \delta^d}\right)$$

Substituting $k = \Theta\left(\ln\left(\frac{1}{\alpha}\right) + \ln\left(N(X,\delta/2)\right)\right) = \Theta\left(\ln\left(\frac{1}{\alpha}\right) + \ln\left(\frac{C_{X,d}}{\delta^d}\right)\right)$:

$$n = \Theta\left(\frac{C_{X,d}}{\delta^d} \cdot \ln\left(\frac{1}{\alpha}\right) + \frac{C_{X,d}}{\delta^d} \cdot \ln\left(\frac{C_{X,d}}{\delta^d}\right)\right)$$

$$n = \Theta\left(\frac{C_{X,d}}{\delta^d} \cdot \ln\left(\frac{1}{\alpha}\right) + \frac{C_{X,d}}{\delta^d} \cdot \left(\ln(C_{X,d}) + d \cdot \ln\left(\frac{1}{\delta}\right)\right)\right)$$

$$n = \Theta\left(\frac{C_{X,d}}{\delta^d} \cdot \ln\left(\frac{1}{\alpha}\right) + \frac{C_{X,d} \cdot d}{\delta^d} \cdot \ln\left(\frac{1}{\delta}\right)\right)$$

For fixed $\delta$ and decreasing $\alpha$, the dominant term is $\Theta\left(\frac{C_{X,d}}{\delta^d} \cdot \ln\left(\frac{1}{\alpha}\right)\right)$.

By the union bound, the probability that any region has fewer than $k$ samples is at most $\alpha$. This ensures that with probability at least $1 - \alpha$, every query point has at least $k$ neighbors within distance $\delta$.

**Step 4:** We connect this to the regret bound.

With probability at least $1 - \alpha$, every query has at least $k$ neighbors within distance $\delta$. For such queries, by the $\delta$-locality property, the kNN router achieves regret $O(\epsilon(\delta))$ since: $|\hat{u}(x,m) - u(x,m)| < \epsilon(\delta)$ for all models $m$.

When selecting the model with the highest predicted score:

$$m_{kNN} = \arg\max_m \hat{u}(x,m),$$

we have: $u(x,m_{kNN}) > \hat{u}(x,m_{kNN}) - \epsilon(\delta) \leq \hat{u}(x,m*) - \epsilon(\delta) > u(x,m*) - 2\epsilon(\delta)$.

Therefore, the regret is bounded by:

$$u(x,m*) - u(x,m_{kNN}) < 2\epsilon(\delta)$$

**Part 2: Parametric Router Sample Complexity** For a parametric router with $L$ Lipschitz-continuous layers, we analyze the sample complexity required to learn an accurate model of the performance function.

**Step 1:** We establish the approximation capacity.

Following the results of (Barron, 1993) and (Yarotsky, 2017), a neural network with $L$ layers and width $W$ can approximate functions in certain smoothness classes to accuracy $\epsilon$ if $W = \Omega((1/\epsilon)^{d/L})$, where $d$ is the input dimension.

For the utility function $u(x,m)$, which maps from $\mathbb{R}^d \times \mathcal{M}$ to $\mathbb{R}$, a network with $L$ Lipschitz-continuous layers requires $\Omega(L \cdot (1/\epsilon)^{d/L})$ parameters to achieve uniform approximation error at most $\epsilon$.

**Step 2:** We relate approximation capacity to sample complexity.

By standard generalization bounds for neural networks (Bartlett et al., 2017; Golowich et al., 2018), the sample complexity to learn the parameters of such a network with generalization error at most $\epsilon$ is:

$$n_{\text{param}} = \Omega\left(\frac{W \cdot L \cdot \log(W)}{\epsilon^2}\right)$$

Substituting $W = \Omega(L \cdot (1/\epsilon)^{d/L})$, we get:

$$n_{\text{param}} = \Omega\left(\frac{L^2 \cdot (1/\epsilon)^{d/L} \cdot \log(L \cdot (1/\epsilon)^{d/L})}{\epsilon^2}\right)$$

For small values of $\epsilon$ and moderate values of $L$, the dominant term is $\Omega(L/\epsilon^2)$, which represents a lower bound on the sample complexity.

**Step 3:** We connect to the regret bound.

To achieve a regret bound of $O(\epsilon(\delta))$, the parametric model must approximate the true performance function with error at most $\epsilon(\delta)/2$ uniformly over the query space. This requires a sample complexity of $\Omega(L/\epsilon(\delta)^2)$.

**Part 3: Comparison** Now we compare the sample complexity of the two approaches:

kNN router: $\Theta\left(\frac{C_{\mathcal{X},d}}{\delta^d} \cdot \log\left(\frac{1}{\alpha}\right)\right)$

Parametric router: $\Omega(L/\epsilon(\delta)^2)$

For small values of $\epsilon(\delta)$ (high accuracy requirements), the parametric router's sample complexity grows quadratically with $1/\epsilon(\delta)$, while the kNN router's complexity depends on $1/\delta^d$ and only logarithmically on $1/\alpha$.

When the embedding space has a low intrinsic dimension $d$ (which is often the case for well-designed embedding spaces), and $\epsilon(\delta)$ decreases rapidly with $\delta$ (strong locality property), the kNN router requires significantly fewer training samples than a parametric router to achieve the same regret bound.

$\square$