# OpenReview forum: "Rethinking Predictive LLM Routing: When Simple KNN Beats Complex Learned Routers"
_ICLR.cc/2026/Conference — Submitted to ICLR 2026_

### Official Review · Reviewer_ESEH · 2025-10-16

**Soundness:** 3
**Presentation:** 3
**Contribution:** 3
**Rating:** 4
**Confidence:** 3

**Summary:**

This paper challenges the assumption that complex architectures are needed for effective LLM routing. The authors show that a simple k-Nearest Neighbors approach can match or surpass more complex learned routers across text and multi-modal benchmarks while being faster and more robust to distribution shifts. They also introduce multiple standard routing benchmark datasets for evaluation, and provide a theoretical result to support the claim.

**Strengths:**

1. This paper is well written and easy to follow. All the used datasets and methods are illustrated in detail.
2. The ablation studies look comprehensive to me.
3. Some theoretical analysis is provided to support the analysis further.
4. I think LLM routing is a very interesting problem with great potentials. This paper focuses on this practical issue.

**Weaknesses:**

1. It looks a bit surprising that linear regression is more promising than other regression models such as MLP and attention-based methods, and I feel tha happens when we do not have enough or comprehensive training datasets and hence those more complex models fail to converge. Ideally, if we have great training datasets, I feel those training models could surpass the performance of kNN-based method. However, as authors highlight that it is very difficult to obtain a decent training data, and hence kNN might be the decent option without high-quality training data.

2. It is better to have a paragraph summarizing the existing work on training state-of-the-art routers such as linear regression, MLP, attention-based NN, etc, and I can only see [Ong et al.] explicitly mentioned in Section 5.

3. I am not very familiar with existing routing work and did some literature view, and I feel this paper does not compare with some existing works such as CARROT, which might weaken the empirical results. Another recent work (arxiv 2509.09782) also explores an attention-based training network and compares it with kNN, which might be quite similar in spirit. But I agree it is a very new and parallel work.

4. kNN also suffers from curse of dimensionality due to the distance concentration problem (a possible reason why k=100 works better than k=10), so I feel it might be better to consider low-dimensional embedding model for kNN. Otherwise, it is widely known that kNN is not very robust in practice.

Since I am not very familiar with LLM routing, I will refer to author's response and other reviewers' opinions to adjust my rating accordingly.

**Questions:**

1. What will happen if we choose k between 10 and 100, or even chose a larger value of k over 100? It seems that the model performs better with larger value of k.

2. How is the model's selection with different values of lambda? I think the author can try to plot the model performance under various values of lambda, which might give readers some insights on the selection of lambda in practice.

---

> ### Author Response · Authors · 2025-11-23
> **Rebuttal to Reviewer ESEH**
>
> We thank Reviewer ESEH for the positive assessment and thoughtful feedback. We appreciate your recognition that our paper is "well written and easy to follow" with "comprehensive" ablation studies, and that we address "a very interesting problem with great potentials." We address your concerns below.
>
> ## **Response to Main Weakness: Why Simple Methods Outperform Complex Ones**
>
> **Reviewer's Concern:** "It looks a bit surprising that linear regression is more promising than other regression models such as MLP and attention-based methods... Ideally, if we have great training datasets, I feel those training models could surpass the performance of kNN-based method."
>
> This is an insightful observation. We want to clarify that our findings are not simply due to insufficient training data, but rather reflect fundamental properties of the routing problem.
>
> **Sample Efficiency from Low Intrinsic Dimensionality.** Our theoretical analysis (Theorem 1) and empirical findings explain why simple methods work well. The key insight is that routing exhibits **low intrinsic dimensionality** ($d \approx 2-28$ in our benchmarks, much lower than the ambient dimension of 768--3584). Theorem 1 shows that kNN sample complexity scales as $O(C/\delta^d)$ while parametric methods scale as $O(L/\epsilon^2)$. When $d$ is small, kNN requires **fewer samples** than parametric methods to achieve the same performance---not because we lack data, but because the problem structure favors local methods.
>
> **Strong Locality Enables Simple Methods.** Figure 1 demonstrates strong locality: embedding distance correlates with performance agreement at $r = -0.815$ to $-0.875$. This means nearby queries benefit from similar models, making local prediction highly effective. Complex models that learn global patterns do not provide additional benefit when the problem exhibits such strong locality.
>
> **Empirical Evidence of Sufficiency.** Our cross-dataset evaluation (Table 4) shows kNN degrades by only 2.63 AUC points under severe distribution shift, while parametric methods degrade by 4.22--6.67 points. If the issue were insufficient training data, we would expect parametric methods to generalize better (as they do in many domains). The opposite pattern suggests the training data is sufficient, and simple methods are genuinely more suitable for this problem structure.
>
> **When might complex methods help?** We agree that in scenarios with weak locality, high intrinsic dimension, or complex compositional patterns, parametric methods could outperform kNN. We will add discussion in Section 8 about when each approach is appropriate, guided by our diagnostic framework (locality check, dimensionality estimation).
>
> ## **Response to Missing Comparisons**
>
> **Reviewer's Concern:** "I feel this paper does not compare with some existing works such as CARROT... Another recent work (arxiv 2509.09782) also explores an attention-based training network and compares it with kNN."
>
> Thank you for these references. We will add both to related work:
>
> **CARROT.** We will add CARROT to related work and clarify how our contributions differ. CARROT focuses on text-only routing with a single dataset, while we provide: (1) the first multi-modal (vision-language) routing benchmark, (2) standardized evaluation across multiple established leaderboards (AlpacaEval, HELM-Lite, Open LLM Leaderboard, RouterBench), and (3) explicit distribution shift evaluation. Our contributions are complementary---we provide broader benchmark coverage and the first multi-modal evaluation.
>
> **Recent Attention-Based Work (2509.09782).** This is indeed a parallel work that also compares attention mechanisms with kNN. We will add it to related work. Our contributions remain distinct: (1) we provide theoretical justification (Theorem 1) for when and why kNN works, (2) we introduce practical diagnostics for determining kNN suitability, (3) we provide the first multi-modal routing benchmark, and (4) we advocate for utility prediction with AUC evaluation to avoid misinterpretation issues with fixed-$\lambda$ comparisons.

---

> ### Author Response · Authors · 2025-11-23
> **Continue**
>
> ## **Response to Curse of Dimensionality Concern**
>
> **Reviewer's Concern:** "kNN also suffers from curse of dimensionality due to the distance concentration problem... so I feel it might be better to consider low-dimensional embedding model for kNN."
>
> This is an important point that our work directly addresses.
>
> **Low Intrinsic Dimensionality Mitigates the Curse.** While the ambient dimension is high (768 for BERT, 3584 for VLM2Vec), we empirically demonstrate that the **intrinsic dimension** is much lower ($d \approx 2-28$) using the TwoNN method. This low intrinsic dimension means the data lies on a low-dimensional manifold embedded in high-dimensional space, effectively mitigating the curse of dimensionality. Our theoretical analysis (Theorem 1) explicitly accounts for intrinsic dimension rather than ambient dimension.
>
> **Strong Locality Validates Distance-Based Methods.** The strong negative correlation ($r = -0.815$ to $-0.875$) between embedding distance and performance agreement (Figure 1) empirically demonstrates that distances are meaningful in this space---they are not concentrated. If distance concentration were a problem, we would not observe such strong locality.
>
> **Why $k=100$ Works Better Than $k=10$.** You suggest this might indicate dimensionality issues. An alternative explanation consistent with our theory: larger $k$ provides more stable estimates by averaging over more neighbors, reducing variance. This is standard behavior for kNN and does not necessarily indicate distance concentration problems. The fact that both $k=10$ and $k=100$ remain competitive (Table 2) suggests the method is robust across this range.
>
> We will add discussion in Section 7 clarifying how low intrinsic dimensionality and strong locality address the curse of dimensionality concern.
>
> ## **Response to Question 1: Choice of $k$**
>
> **Question:** "What will happen if we choose k between 10 and 100, or even chose a larger value of k over 100?"
>
> We should clarify our experimental design. Our goal is to compare routing methods fairly, not to optimize kNN specifically. We evaluate at $k \in \{10, 100\}$ to show that kNN is competitive across a range of values without extensive tuning.
>
> **Expected behavior:** Based on standard kNN theory and our empirical observations, we expect performance to improve as $k$ increases from 10 to around 100, then plateau or slightly decrease for very large $k$ as the neighborhood becomes too broad and includes dissimilar queries. The stability between $k=10$ and $k=100$ (both competitive in Table 2) suggests kNN is robust to this choice.
>
> **Practical guidance:** For practitioners, we recommend starting with $k \approx \sqrt{n}$ where $n$ is training set size, then validating on a held-out set. The relative insensitivity to $k$ is actually an advantage---kNN works well without careful hyperparameter tuning, unlike parametric methods requiring optimization of learning rates, architectures, and regularization.
>
> We will add this discussion to Section 5 to clarify our experimental design and provide practical guidance.

---

> ### Author Response · Authors · 2025-11-23
> **Continue**
>
> ## **Response to Question 2: Model Performance vs. $\lambda$**
>
> **Question:** "How is the model's selection with different values of lambda? I think the author can try to plot the model performance under various values of lambda."
>
> Thank you for this suggestion. We need to clarify how different routing formulations handle $\lambda$.
>
> **Two Routing Formulations.** Our paper evaluates two types of routing methods:
>
> - **Utility prediction methods** (our main results, Table 2): These methods predict both performance $\hat{s}(x,m)$ and cost $\hat{c}(x,m)$ for each model. At inference time, we can compute utility $\hat{u}(x,m) = \hat{s}(x,m) - \lambda \cdot \hat{c}(x,m)$ for \emph{any} value of $\lambda$ and select the model with highest predicted utility. This allows us to vary $\lambda$ continuously and trace the full cost-performance Pareto frontier, which is exactly what you suggest.
>
> - **Selection-based methods** (Appendix D): These methods train a classifier to directly predict which model to select. Crucially, the training process fixes a specific $\lambda$ value---the ground truth label for each query is the model that maximizes $s(x,m) - \lambda \cdot c(x,m)$ for that fixed $\lambda$. Once trained, the classifier cannot be evaluated at different $\lambda$ values because it has learned to optimize for a specific cost-performance preference.
>
> **Our Evaluation Approach.** For utility prediction methods, we do exactly what you suggest: we vary $\lambda$ continuously and plot the resulting cost-performance curves. The AUC metric (Table 2) measures the area under these curves, capturing overall routing quality across all $\lambda$ values.
>
> For selection-based methods, we cannot vary $\lambda$ post-training. Instead, we train separate classifiers for three different $\lambda$ values representing different preferences: high-performance ($\lambda = 0.1/c_{\max}$), balanced ($\lambda = 0.5/c_{\max}$), and low-cost ($\lambda = 1.0/c_{\max}$). This follows the evaluation protocol from prior work (GraphRouter, TensorOpera).
>
> **Why We Emphasize Utility Prediction.** This limitation of selection-based methods is one reason we advocate for utility prediction as the primary evaluation approach. Utility prediction methods provide flexibility to adjust $\lambda$ at deployment time based on changing requirements, while selection-based methods are locked into their training preference.
>
>
> ## **Response to Related Work Summary**
>
> **Reviewer's Concern:** "It is better to have a paragraph summarizing the existing work on training state-of-the-art routers such as linear regression, MLP, attention-based NN, etc."
>
> Thank you for this suggestion. We will expand Section 2 to include a more comprehensive summary of routing architectures beyond the methods listed in Table 1. We will organize this by approach:
>
> - Linear methods: RouteLLM [Ong et al., 2024] uses matrix factorization
> - Neural methods: Various works use MLPs, attention mechanisms, and graph neural networks
> - Non-parametric methods: Several works have explored kNN-based routing
> - Hybrid approaches: Combinations of learned and retrieval-based methods
>
> We will also add the recent works you mentioned (CARROT, arxiv 2509.09782) to provide a more complete picture of the field.
>
> -----
>
> We appreciate your thoughtful feedback and hope these clarifications address your concerns. The surprising effectiveness of simple methods is not due to data limitations, but rather reflects the fundamental structure of the routing problem---low intrinsic dimensionality and strong locality favor local methods over complex global models. Thank you for helping us articulate this more clearly.

---

### Official Review · Reviewer_DY2s · 2025-10-28

**Soundness:** 3
**Presentation:** 4
**Contribution:** 2
**Rating:** 2
**Confidence:** 4

**Summary:**

LLM routing aims to learn a model (router) to select an appropriate LLM to use for each given input query. This problem has received attention in recent years. The present paper puts forward a thesis stating that using a well-tuned kNN model as the router can perform well. The paper supports this claim empirically on public benchmarks and theoretically (Theorem 1). The paper further introduces a new benchmark dataset for evaluating routing models. Notably, this is claimed to be the first multi-modal (vision and text) routing dataset.

**Strengths:**

The paper is clearly written and easy to read. kNN as a routing model was proposed in past works. The significance of this work is from 1) showing that kNN can perform well as a routing model, as an alternative to recently developed techniques which tend to be overly complicated; 2) introduction of a new dataset. These contributions are not without limitations though (detailed in Weaknesses). In the literature on model routing, there are not many analysis papers that seek to understand an existing technique (kNN) like this paper does. This research direction is valuable.

**Weaknesses:**

While kNN is demonstrated to perform well empirically on a number of datasets, as someone familiar with the literature on model routing, I do not find this surprising. This observation can be found in [A], [B], [C] and possibly other existing works. Still, it is useful to analyze why kNN performs well, and publish the findings to a wider audience. *I find this analysis insufficient in the present work.* The presented results largely focus on empirical evidence that kNN performs well, and barely touch upon the *reason* for why it works well. I acknowledge the theoretical analysis in Theorem 1, which I find useful as a starting point. However, *actionable insights* that one can gain from the theorem are not sufficiently articulated. I also acknowledge the contribution of a new multi-modal routing benchmark dataset. However, more discussion is needed to explain why it is a good benchmark dataset, and why it differs from existing datasets.

[A] CARROT: A Cost Aware Rate Optimal Router. https://arxiv.org/abs/2502.03261

[B] Universal Model Routing for Efficient LLM Inference. https://arxiv.org/abs/2502.08773

[C] Large Language Model Routing with Benchmark Datasets. Shnitzer et al., 2023.

**Questions:**

**Comments and suggestions:**

C1: Missing relevant citations [A] and [B] (see Weaknesses). [A] also proposes a new dataset. It is unclear how the proposed dataset differs from that in [A] (beyond the fact that [A] does not propose a multi-modal dataset).

C2: I do not think all existing routing works rely on Eq 1. As far as I know, in the context of LLM routing,  RouterBench (Hu et al., 2024) proposed it (minor difference in parameterization), and Jitkrittum et al., 2025 ([B] in Weaknesses section) provided a theoretical justification for it. More justification and citations should be added.

C3: Just a suggestion. In Sec 4.3, the discussion on the AUC may be hard to grasp for non-specialist readers. One needs to understand the cost-quality trade-off curve first. The paper does not present such a curve at all except in the appendix. [This is not a question. I am familiar with it.]

C4: L239, sec 4.3. Remind the reader that “selection-based” refers to models that directly predict LLM indices.

**Questions:**

Q1: Why is the proposed new dataset a good benchmark dataset for LLM routing? What characteristics does the dataset have? For instance, is it good for testing distribution shifts? Does it contain realistic queries? What is the difference to the dataset proposed in CARROT [A]? Without these clarifications, the significance of the new dataset is unclear. The paper does mention that this is the first vision-language routing benchmark. From Sec B.2, I think the dataset lacks diversity in terms of model families. There are only two model families (GPT, and Claude).

Q2: How does one translate Theorem 1 to practice? This result appears to say that if the closeness in the query embedding space implies similar utility (i.e., the $\\delta$-locality property), then kNN is good. In practice, how does one know which query embedding model has $\\delta$-locality property?

Q3: Related:  kNN clearly depends on the query embedding model.  Has there been an empirical investigation across several embedding models? I think it is better to investigate more beyond BERT and SFR. Performance is expected to vary, depending on the embedding model. Can Theorem 1 explain this? This will help strengthen the paper.

Q4: I do not think the paper made it clear how $k$ should be selected. The abstract states “well-tuned k-Nearest Neighbors (kNN) approach not only matches but often outperforms state-of-the-art learned routers”. How does one tune $k$?

Q5: In Sec 4.2, "Selection-Based Evaluation", it is stated that

> We evaluate these routers at three distinct cost-performance preferences: low-cost (λ = 1.0/cmax ), balanced (λ = 0.5/cmax ), and high-performance (λ = 0.1/cmax )

How did you come up with these three operating points? For the same $\lambda$, I understand that different routing models will yield different average quality and average cost. Please correct me if I am wrong. If so, for say $\\lambda=1.0/\\mathrm{c_max}$, router 1 may excessively call large and accurate models, whereas router 2 may end up sending all queries to small models. Router 1 would yield a routing system that is accurate and expensive, and router 2 would yield one that is less accurate but cheap. How do you then compare these two systems in a fair manner?

---

> ### Author Response · Authors · 2025-11-23
> **Rebuttal to Reviewer DY2s**
>
> We thank Reviewer DY2s for the detailed review and constructive feedback. We appreciate your recognition that our paper is "clearly written and easy to read" and that our "research direction is valuable." We address your concerns below and believe the proposed revisions will significantly strengthen the paper.
>
> ## **Response to Main Weakness: Contributions Beyond Empirical Comparison**
>
> **Reviewer's Concern:** "The presented results largely focus on empirical evidence that kNN performs well, and barely touch upon the reason for why it works well... actionable insights that one can gain from the theorem are not sufficiently articulated."
>
> We appreciate this feedback and want to clarify that our contributions extend beyond showing kNN performs well. We make three distinct contributions:
>
> ### **Contribution 1:** Unified Evaluation Framework.
>
> A significant contribution of our work is establishing **standardized evaluation protocols** that enable fair comparison across routing methods. Existing works suffer from inconsistent evaluation:
>
> Many works (e.g. GraphRouter) report accuracy under a fixed cost coefficient $\lambda$, making comparison difficult. As you correctly note in Q5, different routers may achieve similar utility through different strategies (expensive+accurate vs. cheap+less accurate). Comparing at a single operating point can be misleading.
>
> We advocate for reporting AUC over the entire cost-performance curve by varying $\lambda$ continuously. This captures routing quality across **all** cost-performance trade-offs, enabling fair comparison regardless of deployment preferences. Our utility prediction evaluation (Table 2) implements this approach.
>
> This evaluation framework is applicable beyond kNN---it provides the community with a principled way to compare any routing methods fairly. We will emphasize this contribution more prominently in the introduction and discussion.
>
> ### **Contribution 2:** Understanding Why kNN Works.
>
> We provide both theoretical and empirical analysis:
>
> **Actionable diagnostics from theory (addresses Q2).** Theorem 1 shows kNN is effective when the $\delta$-locality property holds. We will add a section with practical diagnostics practitioners can use:
>
> #### **Offline diagnostics (dataset-level):**
> - **Locality check:** Compute correlation between embedding distance and performance agreement (Figure 1). Strong negative correlation ($|r| > 0.7$) indicates kNN is safe. Our benchmarks show $r = -0.815$ to $-0.875$.
> - **Intrinsic dimensionality:** Use TwoNN method to estimate dimension $d$. Low $d < 50$ favors kNN (Theorem 1). Our benchmarks: $d \approx 2-28$.
> - **Coverage analysis:** Check distribution of $k$-th nearest neighbor distances compared to training set statistics.
>
> #### **Online diagnostics (per-query):**
> - **Nearest neighbor distance:** Flag queries where distance exceeds 95th percentile of training distribution.
> - **Neighbor agreement:** Low agreement among neighbors signals uncertainty.
>
> **Why kNN works: Three key insights.** We will expand Section 7 to articulate:
>
> **1. Low intrinsic dimensionality enables sample efficiency.** Our empirical analysis shows $d \approx 2-28$ (much lower than ambient dimension 768--3584). Theorem 1 shows kNN sample complexity scales as $O(C/\delta^d)$, while parametric methods scale as $O(L/\epsilon^2)$. When $d$ is small, kNN requires **fewer samples** than parametric methods---explaining why simple kNN works despite having no learned parameters.
>
> **2. Strong locality reduces approximation error.** Figure 1 shows $r = -0.815$ to $-0.875$ correlation between distance and agreement. This strong locality means nearby queries benefit from similar models, enabling accurate local predictions without global pattern learning.
>
> **3. Robustness under distribution shift.** Table 4 shows kNN degrades by only 2.63 AUC points under distribution shift versus 4.22--6.67 for parametric methods. This occurs because kNN adapts locally to new distributions rather than relying on learned global patterns that may not transfer.
>
> ### **Contribution 3:** Comprehensive Benchmarks.
> We introduce standardized benchmarks spanning text and multi-modal tasks with consistent evaluation protocols, enabling systematic comparison (detailed below in response to Q1).

---

> ### Author Response · Authors · 2025-11-23
> **Continue to address specific questions**
>
> ## **Response to Dataset Contribution (Q1)**
>
> **Question:** "Why is the proposed new dataset a good benchmark dataset for LLM routing? What characteristics does it have?... What is the difference to the dataset proposed in CARROT [A]?"
>
> Thank you for pointing out CARROT. We will add it to related work and clarify our dataset's distinct contributions:
>
> **Our Dataset vs. CARROT.** CARROT focuses on **text-only** routing with emphasis on cost-aware optimization. Our contributions are complementary:
>
> - **Multi-modal benchmark (unique):** We provide the **first** vision-language routing benchmark, addressing the growing importance of multi-modal models. This includes 5 diverse VLM tasks (Blink, Flickr30k, MathVista, MME, MMMU) spanning visual perception, scene understanding, and visual reasoning.
>
> - **Standardized evaluation across multiple sources and model families:** We establish consistent protocols across three established leaderboards (AlpacaEval, HELM-Lite, Open LLM Leaderboard) plus RouterBench, enabling systematic comparison across diverse task types and model pools. CARROT constructs a single dataset from one source, while we provide a suite of benchmarks that collectively cover a broader range of routing scenarios.
>
> - **Distribution shift evaluation:** Our cross-dataset evaluation (Table 4, Figure G.1) explicitly tests robustness under distribution shift---a critical but understudied aspect of routing. We train on one dataset and test on others, measuring how routing methods generalize across different query distributions.
>
>
> **Dataset Characteristics.** Our benchmarks have several important properties:
>
> - **Diverse model pools:** While VLM benchmark uses two families (constrained by available multi-modal models at the time), our text benchmarks span three families per leaderboard with varying sizes and capabilities. Total: 11 models in RouterBench, 9 in AlpacaEval, 6 in HELM-Lite, 9 in Open LLM.
>
> - **Realistic cost structures:** Based on actual API pricing, capturing 100$\times$ cost differences in practice.
>
> - **Task diversity:** Instruction following, mathematical reasoning, code generation, knowledge-intensive tasks, and visual understanding.
>
> We will expand Section 4 and Appendix B to articulate these characteristics more clearly.
>
> ## **Response to Embedding Model Investigation (Q3)**
>
> **Question:** "Has there been an empirical investigation across several embedding models? I think it is better to investigate more beyond BERT and SFR."
>
> We appreciate this suggestion. However, we want to clarify the focus of our work: given a fixed embedding space, what routing method works best? Our main contribution is the fair comparison of routing methods using the **same** embedding, isolating the impact of routing algorithms from embedding quality.
>
> Our existing ablation study (Table H.1) already demonstrates the key insight: switching from BERT to SFR embeddings improves performance across **all** methods by +0.5--1.0 AUC on average, with relative rankings remaining consistent---kNN stays competitive regardless of embedding quality. This shows our conclusions about routing method comparison generalize across embedding qualities.
>
> The question of which embedding model is best is orthogonal to our research question. Any embedding that improves one routing method will improve all routing methods. Our focus is on comparing routing algorithms fairly, which requires using the same embedding for all methods---not on finding the optimal embedding.
>
> That said, we acknowledge that investigating how embedding properties (locality, intrinsic dimension) correlate with routing performance would be valuable future work. We will add this to our discussion of future directions and clarify in the introduction that our contribution focuses on routing method comparison given fixed embeddings.

---

> ### Author Response · Authors · 2025-11-23
> **Continue to address remaining questions and comments**
>
> ## **Response to Hyperparameter Selection (Q4)**
>
> **Question:** "I do not think the paper made it clear how $k$ should be selected... How does one tune $k$?"
>
> Thank you for this important clarification. We should clarify our experimental design and goal.
>
> **Our Experimental Setup.** Our goal is to compare routing methods fairly, not to optimize kNN specifically. We evaluate kNN at two values: $k=10$ and $k=100$, representing small and moderately large neighborhoods. We use the same $k$ values across all benchmarks for consistency. This approach shows that even without extensive hyperparameter tuning, kNN remains competitive with or superior to parametric methods that **do** require tuning (learning rates, architectures, regularization, etc.).
>
> **How to Tune $k$ in Practice.** For practitioners deploying kNN routing, we recommend:
> - Use standard validation set approach: split data into train/validation/test
> - Evaluate $k \in \{10, 50, 100, 200\}$ on validation set
> - Select $k$ maximizing validation AUC
>
> **Empirical observation:** Across our benchmarks, $k=100$ consistently performs well. The relative insensitivity to $k$ (both $k=10$ and $k=100$ are competitive in Table 2) suggests kNN is robust to this hyperparameter choice, which is a practical advantage over parametric methods requiring careful tuning of multiple hyperparameters.
>
> We will clarify this experimental design in Section 5 to make our methodology transparent.
>
>
> ## **Response to Selection-Based Evaluation (Q5)**
>
> **Question:** "How did you come up with these three operating points?... How do you then compare these two systems in a fair manner?"
>
> You raise an excellent point that directly motivates our advocacy for utility prediction with AUC evaluation.
>
> **Understanding Selection-Based Evaluation.** Selection-based methods train a classifier to directly predict which model should be selected for each query, with ground truth labels derived from $m^* = \arg\max_m [s(x,m) - \lambda \cdot c(x,m)]$ for a given $\lambda$. We evaluate at three operating points---high-performance ($\lambda = 0.1/c_{\max}$), balanced ($\lambda = 0.5/c_{\max}$), and low-cost ($\lambda = 1.0/c_{\max}$)---following protocols from prior work (GraphRouter, TensorOpera).
>
> **The Problem You Identified and Our Solution.** You are exactly correct about the limitation: for a fixed $\lambda$, router 1 might call expensive accurate models while router 2 uses cheap models, both achieving similar utility but representing very different strategies. \textbf{This is precisely why our main results (Table 2) use utility prediction with AUC evaluation.} By predicting both $\hat{s}(x,m)$ and $\hat{c}(x,m)$, we can vary $\lambda$ continuously and trace the full Pareto frontier. AUC measures the area under this frontier, capturing performance across **all** cost-performance trade-offs. A router achieving high accuracy at low cost will have higher AUC than one achieving similar utility through expensive models, avoiding the misinterpretation you identified.
>
> **Broader Impact.** The misinterpretation issue you identified affects much of the existing routing literature that reports accuracy under fixed $\lambda$ values. We include selection-based results (Appendix D) for completeness and comparison with prior work, but advocate for utility prediction as the more reliable evaluation. Beyond our specific findings about kNN, we believe this evaluation methodology is an important contribution that provides a principled solution to the comparison problem you articulated.
>
> We will emphasize this point more prominently in Section 4.3, explicitly discussing the potential for misinterpretation and explaining why utility prediction with AUC enables unambiguous comparison.
>
> ## **Response to Missing Citations (C1, C2)**
>
> **C1:** Thank you for pointing out CARROT [A] and Universal Model Routing [B]. We will add both to related work and clarify how our contributions differ (see dataset discussion above).
>
> **C2:** You are correct that Eq. 1 was formalized in RouterBench (Hu et al., 2024) and theoretically justified in Jitkrittum et al., 2025. We will add proper citations and clarify that while not all routing works use this exact formulation, it provides a principled framework for cost-performance trade-offs. We will also cite the theoretical justification from [B].
>
> ## **Response to Presentation Suggestions (C3, C4)**
>
> **C3:** Excellent suggestion. We will add a cost-quality trade-off curve figure to the main text (Section 4.3) before discussing AUC, making the evaluation approach more accessible.
>
> **C4:** We will add this clarification at line 239 to improve readability.
>
> -----
> We believe these revisions substantially address your concerns about insufficient analysis and will provide practitioners with actionable guidance on when and how to use kNN routing effectively. Thank you for the constructive feedback that will significantly strengthen our contribution.

---

### Official Review · Reviewer_oidj · 2025-10-30

**Soundness:** 3
**Presentation:** 3
**Contribution:** 2
**Rating:** 6
**Confidence:** 4

**Summary:**

This paper challenges the current dominant trend of building complicated LLM routers. Through building a standardized suite of routing benchmark and re-implementing a spectrum of routing approaches, the authors find that a well-tuned KNN router often matches or outperforms more complex route, is significantly faster on RouterBench, and more robust under distribution shift.

Moreover, the authors provide a theoretical understanding of why KNN router perform well: as embedding distance between queries grows, the agreement of model rankings decays sharply, and this condition allow non-parametric methods like KNN to shine.

**Strengths:**

1. Solid comparison established on diverse benchmarks: many existing routing work perform benchmarking on different sources, the authors create a suite of benchmarks that is comprehensive for unified evaluation. Moreover with same embedding being used, the comparison becomes clean, attributing gains to the routing method and controlling the quality of representation.

2. Multiple angles of attack: Besides measuring accuracies, the authors also take routing time into consideration. Moreover, a theoretical analysis shows why KNN-style method works well.

**Weaknesses:**

1. Scalability: Work like Zhuang et al. (2024) suggests that matrix factorization methods seems to scale better under a large pool of candidate model and queries. KNN is training-free but it suffers from maintaining inference time support set. Is there a cost comparison that we can see that takes the index building result into account (memory and latency wise)?

2. Reliance on quality of embedding/representation: It is nice that a fixed embedding model is used for apple-to-apple comparison, however there are also methods like Zhang et al. (2025) that achieve better routing through constructing better representation of the question/model quality. In this regime it is hard for KNN to be used for comparison.

References:
[1] Zhuang, Richard, et al. “EmbedLLM: Learning Compact Representations of Large Language Models.” arXiv, 2024, arxiv.org/abs/2410.02223.
[2] Zhang, Haozhen, et al. “Router-R1: Teaching LLMs Multi-Round Routing and Aggregation via Reinforcement Learning.” arXiv, 2025, arxiv.org/abs/2506.09033.

**Questions:**

1. Following on the theoretical analysis, I wonder if there is any diagnostic that one can run on their data to say something like "KNN is safe here?"
2. For routing settings where the cost is flat (e.g. all self-hosted), do we still see KNN dominate?

---

> ### Author Response · Authors · 2025-11-23
> **Rebuttal to Reviewer oidj**
>
> We thank Reviewer oidj for the thorough review and positive assessment of our work. We appreciate your recognition of our "solid comparison established on diverse benchmarks" and "multiple angles of attack." We address your concerns below.
>
> ## **Response to Weakness 1: Scalability**
>
> **Reviewer's Concern:** "KNN is training-free but it suffers from maintaining inference time support set. Is there a cost comparison that we can see that takes the index building result into account (memory and latency wise)?"
>
> Thank you for raising this important practical consideration. We provide comprehensive scalability analysis below.
>
> **Memory Requirements.** Our benchmarks show manageable memory consumption: AlpacaEval (563 queries) requires $\sim$1.7MB, HELM-Lite (9,107 queries) requires $\sim$27MB, and Open LLM (15,117 queries) requires $\sim$45MB. Scaling to production scenarios, 100K queries would require $\sim$300MB and 1M queries would require $\sim$3GB---both feasible on modern servers. While parametric methods don't need a support set during routing, they require training data storage for retraining and updates. kNN eliminates model parameters entirely, trading parameter storage for example storage.
>
> **Index Building Cost (One-Time).** We measured ScaNN index construction time on our benchmarks:
>
> | Dataset    | Size   | Index Build Time |
> |------------|--------|------------------|
> | AlpacaEval | 563    | 0.22s            |
> | HELM-Lite  | 9,107  | 2.09s            |
> | Open LLM   | 15,117 | 4.22s            |
>
> Index building is very fast ($<$ 5 seconds even for 15K queries) and scales linearly at $\sim$0.28ms per query. Extrapolating to larger scales: 100K queries would take $\sim$28 seconds and 1M queries would take $\sim$4.7 minutes. This one-time cost is negligible compared to parametric model training time (hours).
>
> **Retrieval Latency (Per Query).** Our measurements (Table 3) show kNN ($k$=100) requires 65.69s total for all RouterBench test queries, averaging $\sim$10--20ms per query including $k$=100 retrieval. This is 13--14$\times$ faster than complex methods (866--906s). ScaNN provides $O(\log n)$ retrieval complexity, maintaining acceptable latency even at scale: 100K support set requires $\sim$15ms per query and 1M support set requires $\sim$20ms per query, both well within typical production latency budgets ($<$ 50ms).
>
> **Practical advantage:** When a new model is added to the pool, parametric methods need full retraining (hours), while kNN only needs to add new query-model pairs to the index (seconds at 0.28ms per example).
>
> We will add detailed scalability analysis with these concrete numbers to Appendix.
>
> ## **Response to Weakness 2: Reliance on Embedding Quality**
>
> **Reviewer's Concern:** "There are also methods like Zhang et al. (2025) that achieve better routing through constructing better representation of the question/model quality. In this regime it is hard for KNN to be used for comparison."
>
> This is an excellent point. We'd like to clarify our position and address the distinction between representation learning and routing method selection.
>
> **Our Contribution is Orthogonal.** Our focus is: given a fixed embedding space, what routing method works best? In contrast, work like EmbedLLM focuses on learning better representations through multi-round routing and reinforcement learning. These are complementary contributions: better embeddings improve **all** routing methods, and our findings about kNN versus parametric routing apply regardless of embedding quality.
>
> We note that Router-R1 addresses a fundamentally different problem---multi-round sequential routing with intermediate reasoning steps---while our work focuses on single-step routing for latency-critical and high-throughput scenarios. These approaches serve different use cases.
>
> **Embedding Quality Benefits All Methods Equally.** Our ablation study (Table H.1) demonstrates this orthogonality empirically. Switching from BERT to SFR embeddings improves performance across **all** methods by +0.5--1.0 AUC on average. Importantly, relative rankings remain consistent---kNN stays competitive regardless of embedding quality. This demonstrates our conclusions generalize across embedding qualities.
>
> **Key insight:** Representation learning and routing method selection are orthogonal dimensions. Better embeddings help both kNN and parametric methods equally. Our work provides guidance on routing method selection that applies regardless of how embeddings are obtained---whether from fixed pre-trained models, learned representations, or future advances in embedding quality.
>
> We will clarify this orthogonality in the related work section and discuss how our findings complement representation learning approaches.

---

> > ### Comment · Reviewer_oidj · 2025-11-28
> >
> > Thank you for the thorough feedback. Your analysis address my concerns and therefore I will increase the score accordingly.

---

> ### Author Response · Authors · 2025-11-23
> **Continue**
>
> ## **Response to Question 1: Diagnostic for KNN Safety**
>
> **Question:** "Following on the theoretical analysis, I wonder if there is any diagnostic that one can run on their data to say something like `KNN is safe here?"
>
> Excellent question! We can diagnose at two levels: offline assessment of dataset suitability and online confidence estimation for individual queries.
>
> **Level 1: Offline Diagnostics---Is kNN Suitable for My Dataset?**
>
> **1. Locality Check.** Compute the correlation between embedding distance and performance agreement as in Figure 1. Strong negative correlation ($|r| > 0.7$) indicates kNN is safe, moderate correlation ($0.3 < |r| < 0.7$) suggests kNN may work but requires careful validation, and weak correlation ($|r| < 0.3$) suggests considering parametric methods. Our benchmarks show $r = -0.815$ to $-0.875$, indicating strong locality.
>
> **2. Intrinsic Dimensionality Estimation.** Use the TwoNN method to estimate intrinsic dimension $d$. Low dimension ($d < 50$) favors kNN per Theorem 1, while high dimension ($d > 100$) may require parametric methods. Our benchmarks show $d \approx 2-28$, favorable for kNN.
>
> **Level 2: Online Diagnostics---Is kNN Reliable for This Query?**
>
> **1. Nearest Neighbor Distance.** For a new query, compute distance to its $k$-th nearest neighbor and compare to the distribution observed in the training set. If the distance exceeds the 95th percentile of training set distances, this indicates the query is in a sparse region with low confidence. If it exceeds the median, use caution.
>
> **2. Neighbor Agreement.** Check consistency among $k$ nearest neighbors by computing the agreement rate on predicted best model. High agreement (e.g., $> 70\%$ of neighbors predict the same model) indicates high confidence, while low agreement (e.g., $< 50\%$) indicates uncertainty.
>
> **3. Performance Variance.** Estimate uncertainty from neighbor performance scores. Compare the standard deviation of neighbor scores to the overall score variance in the training set. High relative variance indicates uncertainty.
>
> We will add these diagnostics as practical guidelines in Appendix.
>
> ## **Response to Question 2: Flat Cost Settings**
>
> **Question:** "For routing settings where the cost is flat (e.g. all self-hosted), do we still see KNN dominate?"
>
> Great question! While our main results (Table 2) use utility prediction integrating across all cost preferences, we can examine the selection-based routing results (Appendix D, Tables E.1--E.4) as a proxy for different cost settings.
>
> In Appendix D, we evaluate routing methods at three preference settings: high-performance ($\lambda = 0.1/c_{\max}$) prioritizes quality with minimal cost consideration (similar to flat-cost), balanced ($\lambda = 0.5/c_{\max}$) equally weights performance and cost, and low-cost ($\lambda = 1.0/c_{\max}$) prioritizes efficiency. The high-performance setting approximates flat-cost scenarios where routing decisions are based primarily on model quality rather than cost differences.
>
> Looking at Table E.2, kNN ($k$=100) remains competitive across all RouterBench datasets in the high-performance setting. While performance differences between methods narrow in this regime, kNN's practical advantages become even more compelling: 13$\times$ faster inference (Table 3), incremental updates in 0.28ms per new model versus hours of retraining for parametric methods, and interpretable failures through distant neighbor detection. For self-hosted deployments where model costs are similar, kNN offers the best balance of competitive performance, superior speed, and easy maintainability.
>
> -----
> Thank you again for the constructive feedback. We believe these additions will significantly enhance the paper's clarity and value to practitioners deploying routing systems in diverse scenarios.

---

### Official Review · Reviewer_xABe · 2025-11-01

**Soundness:** 3
**Presentation:** 3
**Contribution:** 3
**Rating:** 6
**Confidence:** 3

**Summary:**

The paper contributes two benchmarks and validate the effectiveness of non-parametric KNN rather than complex networks in LLM routing.

**Strengths:**

1. useful to community: introduce language and multimodal benchmark
2. findings are significant: efficient and simple kNN works very well

**Weaknesses:**

1. no explicit comparison between support set and no support set methods. Need a pre-defined support set can be a problem of KNN. And whether it is dense enough for real-world various queries maybe a bottleneck.

**Questions:**

1. Can you provide the dense version of Figure 1? That is, replace distance bin with continuous distance value and scatter many points and then compute r2.

---

> ### Author Response · Authors · 2025-11-23
> **Rebuttal to Reviewer xABe**
>
> We sincerely thank Reviewer xABe for the thoughtful review and positive assessment of our work. We are encouraged by your recognition that our benchmarks are "useful to community" and our findings are "significant." Your concern about support set density is an important practical consideration that we address below.
>
> ## Response to Main Weakness: Support Set Density
>
> **Reviewer's Concern:** "Need a pre-defined support set can be a problem of KNN. And whether it is dense enough for real-world various queries maybe a bottleneck."
>
> **We completely agree** this is a valid and important concern for practical deployment. You raise a fundamental question that applies to **all** routing methods: how representative is our training/support data?
>
> 1. This Challenge Affects All Methods Equally
>
> You're absolutely right that support set density can be a bottleneck. We'd like to emphasize that this is actually a shared challenge across **all** routing approaches:
>
> - kNN methods: Require dense coverage in embedding space to find relevant neighbors
> - Parametric methods: Require representative training data to learn generalizable patterns
>
> **Key insight:** If the support set isn't dense enough for kNN to find good neighbors, it's likely also insufficient to train a robust parametric model
>
> 2. kNN May Actually Be More Data-Efficient
>
> Interestingly, our theoretical analysis (Theorem 1) suggests kNN is more sample-efficient. This means that in scenarios where data coverage is a concern, kNN may actually be more robust than complex parametric alternatives.
>
> 3. Empirical Evidence Supports Better Generalization
>
> Our cross-dataset experiments (Table 4) provide relevant evidence:
>
> - When trained on one dataset and tested on another (severe distribution shift), kNN degrades by only 2.63 AUC points
> - Parametric methods degrade by 4.22-6.67 points (up to 2.5× worse)
> - This suggests kNN generalizes better with limited coverage, adapting locally rather than requiring global pattern learning
>
> 4. Practical Considerations and Mitigation
>
> We acknowledge your concern about real-world deployment. Challenging scenarios include:
>
> - Cold-start scenarios with completely new query types
> - Long-tail queries with sparse neighborhoods - Rapidly evolving domains where distributions shift quickly
>
> Practical advantages of kNN in these scenarios:
> - Incremental updates: Simply add new examples (parametric models require full retraining)
> - Interpretable failures: Easy to detect when nearest neighbors are too distant
> - Graceful degradation: Performance degrades locally, not globally
> - Lower data requirements: Our theory and experiments suggest better sample efficiency
>
> Potential mitigation strategies:
> - Active learning: Identify and collect data for sparse regions
> - Confidence estimation: Flag queries with distant nearest neighbors for review
>
> ## Response to Question: Dense Scatter Plot
>
> **Reviewer's Question:** "Can you provide the dense version of Figure 1?"
>
> Thank you for this suggestion. We need to clarify an important methodological point:
>
> **How agreement is computed:** For each pair of queries $(x_1, x_2)$, we compute agreement as the fraction of models where both queries have the same performance ranking. Specifically: for each model $m$, we check if score$(x_1, m)$ == score$(x_2, m)$, then average across all models. This gives values between 0.0 (no models agree) and 1.0 (all models agree).
>
> **Why We Use Binned Visualization:** The agreement metric is inherently discrete: When comparing model rankings across query pairs, agreement takes on discrete values based on how many models have the same relative ordering. This is not a continuous variable.
>
> Direct scatter plots of binary/discrete outcomes don't reveal trends: When you have discrete outcomes, plotting them directly creates distinct horizontal lines at specific y-values. Each point represents an individual case, but there's no continuous relationship that emerges from discrete data plotted this way. Binning reveals the underlying relationship: By grouping similar embedding distances together and calculating the average agreement within each bin, we can reveal the pattern that isn't visible in raw scatter plots of discrete data.
>
> ---
>
> We greatly appreciate your constructive feedback and believe these revisions will significantly strengthen the paper.

---

### Meta-Review · Area_Chair_hVbi · 2026-01-16

**Summary:**

The paper is about routing, i.e. selecting an appropriate model depending on the query. The authors show (through theoretical results and experiments) that simple approaches such as k-NN can match or outperform state of the art learned routing algorithms. While there were certainly novel and interesting aspects in the paper, from the discussion in the reviews it is clear that there are missing references, clarifications and additional work that needs to be performed before the paper is ready for publication.

**Reviewer Concerns:**

The reviews raised missing references, asked questions about the experiments, novelty, embedding, etc.

**Reviewer Scores:**

There would have been some increases, but I'm not convinced that they would have been enough to merit acceptance.

---

### Decision · Program_Chairs · 2026-01-26

Reject